Corrected: Author correction

# Host-associated niche metabolism controls enteric infection through fine-tuning the regulation of type 3 secretion

James P.R. Connolly [1], Sabrina L. Slater[2], Nicky O'Boyle[1], Robert J. Goldstone[3], Valerie F. Crepin[2], David Ruano-Gallego [2], Pawel Herzyk[4], David G.E. Smith [3], Gillian R. Douce[1], Gad Frankel [2] & Andrew J. Roe[1]

Niche-adaptation of a bacterial pathogen hinges on the ability to recognize the complexity of signals from the environment and integrate that information with the regulation of genes critical for infection. Here we report the transcriptome of the attaching and effacing pathogen *Citrobacter rodentium* during infection of its natural murine host. Pathogen gene expression in vivo was heavily biased towards the virulence factor repertoire and was found to be co-ordinated uniquely in response to the host. Concordantly, we identified the host-specific induction of a metabolic pathway that overlapped with the regulation of virulence. The essential type 3 secretion system and an associated suite of distinct effectors were found to be modulated co-ordinately through a unique mechanism involving metabolism of microbiota-derived 1,2-propanediol, which dictated the ability to colonize the host effectively. This study provides novel insights into how host-specific metabolic adaptation acts as a cue to fine-tune virulence.

[1] Institute of Infection, Immunity and Inflammation, University of Glasgow, Glasgow G12 8TA, UK. [2] MRC Centre for Molecular Bacteriology and Infection, Department of Life Sciences, Imperial College, London SW7 2AZ, UK. [3] School of Life Science, Heriot-Watt University, Edinburgh EH14 4AS, UK. [4] Glasgow Polyomics, Wolfson Wohl Cancer Research Centre, University of Glasgow, Garscube Estate, Glasgow G61 1QH, UK. Correspondence and requests for materials should be addressed to J.P.R.C. (email: James.Connolly@glasgow.ac.uk) or to A.J.R. (email: Andrew.Roe@glasgow.ac.uk)

The basic view of bacterial pathogenesis is the ability of an invader to overcome innate host defences using an array of virulence determinants in an 'arms race' between the two kingdoms[1]. In reality, this simplistic definition is complicated by several factors that are essential to the outcome of this process. Pathogens must first recognize their environment before the decision can be made to unleash their arsenal of virulence factors. Moreover, the specific response of such pathogens to the myriad of metabolites and signals derived from both the host itself and the resident microbiota must be integrated with global gene regulation in order to recognize a precise niche and maximize competitiveness therein[2]. This creates a complex interplay between the invader, the microbiota and the host, leaving many factors in the definition of host–pathogen interactions unclear.

*Citrobacter rodentium* is a Gram-negative pathogen that causes transmissible colonic crypt hyperplasia in the large bowel of a murine host[3–5]. It is a member of the attaching and effacing (A/E) family of bacterial pathogens, which includes the enterohaemorrhagic and enteropathogenic *E. coli* pathotypes (EHEC and EPEC, respectively) that are responsible for severe diarrhoeal illness and mortality in humans worldwide[6–8]. EHEC and EPEC do not naturally colonize mice and therefore *C. rodentium* has been adopted as the relevant surrogate model to study their pathogenesis in vivo[9,10].

Genetically, A/E pathogens are defined by the presence of a large ~35 Kb pathogenicity island carried on their chromosome known as the locus of enterocyte effacement (LEE)[11–13]. The LEE comprises 41 genes, largely across five polycistronic operons, and encodes all the necessary components of a type 3 secretion system (T3SS) as well as master regulators, a selection of effector proteins and an adhesin known as Intimin[14]. This T3SS is essential for the pathogenesis of these organisms, facilitating intimate attachment to the host cell surface through the translocation of Tir (translocated Intimin receptor) and the formation of characteristic A/E lesions on the host cell surface[8,14–18]. Furthermore, the T3SS effector protein repertoire of A/E pathogens is vast and extends beyond that encoded within the LEE[19–21]. The roles of effector proteins range from reorganization of host cell actin, effacement of the microvilli around A/E lesions and disruption of tight junctions to immune response modulation and inhibition of apoptosis[7,22].

The T3SS does not rely on a tissue-receptor molecule to mediate attachment and therefore must be tightly regulated in order to ensure appropriate expression and, in turn, niche-specification. Regulation of the LEE is an extremely complex process involving interplay between regulators encoded within the island and globally on the chromosome[14,23,24]. Ultimately, LEE regulation centres on temporal expression of the T3SS in response to the multitude of signals and cues that are encountered in the environment—such as nutrients, pH, oxygen, quorum sensing molecules and host hormones. Moreover, recent work has demonstrated the importance of host cell attachment as a mechanical cue that leads to both transcriptional and post-transcriptional regulatory mechanisms[25,26]. It is the sensing and integration of these signals that is believed to determine to the niche-specific nature of A/E pathogens as opposed to a specific tissue-receptor tropism. However, our current vision is far from complete because most studies rely on in vitro systems that lack important host components that are likely to be key determinants in affecting the dynamics of an infection.

To address this problem, we have probed the transcriptome of *C. rodentium* during infection of its natural host and discovered the expression of a specific metabolic pathway that was induced in response to murine colonization. Metabolism of microbiota-derived 1,2-propanediol enhanced *C. rodentium* fitness in vivo through a fine-tuned co-ordination of T3SS and effector expression, thus contributing to niche-specification within the murine gut. This study has identified a novel regulatory mechanism of colonization and reveals important insights into adaption of a pathogen to a specific host.

## Results

**C. rodentium model of attaching and effacing pathogenesis.** The natural host range of *C. rodentium* combined with its genetic makeup has led to extensive use as a surrogate model of EHEC and EPEC infection[3,10]. In order to gain insights into the specific mechanisms used by *C. rodentium* in vivo we performed RNA-seq using colonized tissue isolated directly during infection. Mice were orally gavaged with $\sim3\times10^9$ CFU of *lux* marked WT *C. rodentium*. Colonization of mice was monitored in real-time using the in vivo imaging system (IVIS) at regular intervals to track the infection (Fig. 1a). Early colonization was detectable at day 3 post infection (PI) with measurable luminescence flux emitting from infected mice being $>1\times10^7$ relative luminescence units (RLU; photons/s/cm$^2$/square-radian). The peak of infection was at day 7 PI with flux emitting at $\sim2$–$4\times10^7$ RLU and the infection decline began at day 10 PI to measurable flux similar to that of day 3 PI (Fig. 1b). No luminescence was detectable in the uninfected control mice (Supplementary Figure 1a).

In order to investigate gene expression dynamics during infection RNA was extracted from infected tissue identified by IVIS—the caecal patch and the terminal rectum (referred to as the caecum and rectum herein). Triplicate mice were sacrificed at each timepoint and small ~5 mm biopsies corresponding to the luminescence signal were immediately dissected, gently cleared of luminal content and used for RNA extraction (Fig. 1c). As a negative control the corresponding regions of uninfected mice were also isolated (Supplementary Figure 1a). *C. rodentium* pathogenesis is known to rely upon the LEE-encoded T3SS[18]. To confirm this, we also gavaged mice with a Δ*ler* mutant strain. Ler is the master regulator of the LEE and as such Δ*ler* is avirulent, resulting in no colonization (Supplementary Figure 1a). DMEM is routinely used to induce the T3SS optimally in vitro, confirmed by profiling the secretome of *C. rodentium* using SDS-PAGE[14]. As expected, known secreted proteins Tir, EspB/D and EspA were all identified in the supernatant of DMEM cultures whereas Δ*ler* did not secrete any effectors (Supplementary Figure 1b). In contrast, bacterial supernatants from suboptimal LEE-inducing conditions (LB) contained no detectable effectors. RNA was thus isolated from cultures of *C. rodentium* grown in DMEM (T3SS +ve) and subjected to RNA-seq to act as a comparator to in vivo derived RNA (Fig. 1d; Supplementary Figure 1c).

**The transcriptome of C. rodentium during murine infection.** We performed in vivo RNA-seq to decipher changes in gene expression of *C. rodentium* during murine infection. Enriched bacterial mRNA was library prepared from infected tissue biopsies and sequenced on the Illumina NextSeq 500 platform. Sequencing obtained between 26 and 45 million reads per sample and quality controlled reads were mapped with high stringency to the *C. rodentium* reference genome (Supplementary Data 1)[27]. This approach allowed robust profiling of transcript levels across the genome throughout infection (Supplementary Figure 2a). Moreover, correlation analysis of replicate samples was strong ($r > 0.8$) further increasing the confidence of the data generated (Supplementary Figure 2b).

Analysis of differentially expressed genes (DEGs) identified by in vivo RNA-seq compared to in vitro conditions (FDR corrected *P*-value $\leq 0.05$) allowed profiling true gene expression dynamics of this pathogen during infection (Supplementary Data 2). We identified 31 DEGs during early colonization (18 up; 13 down),

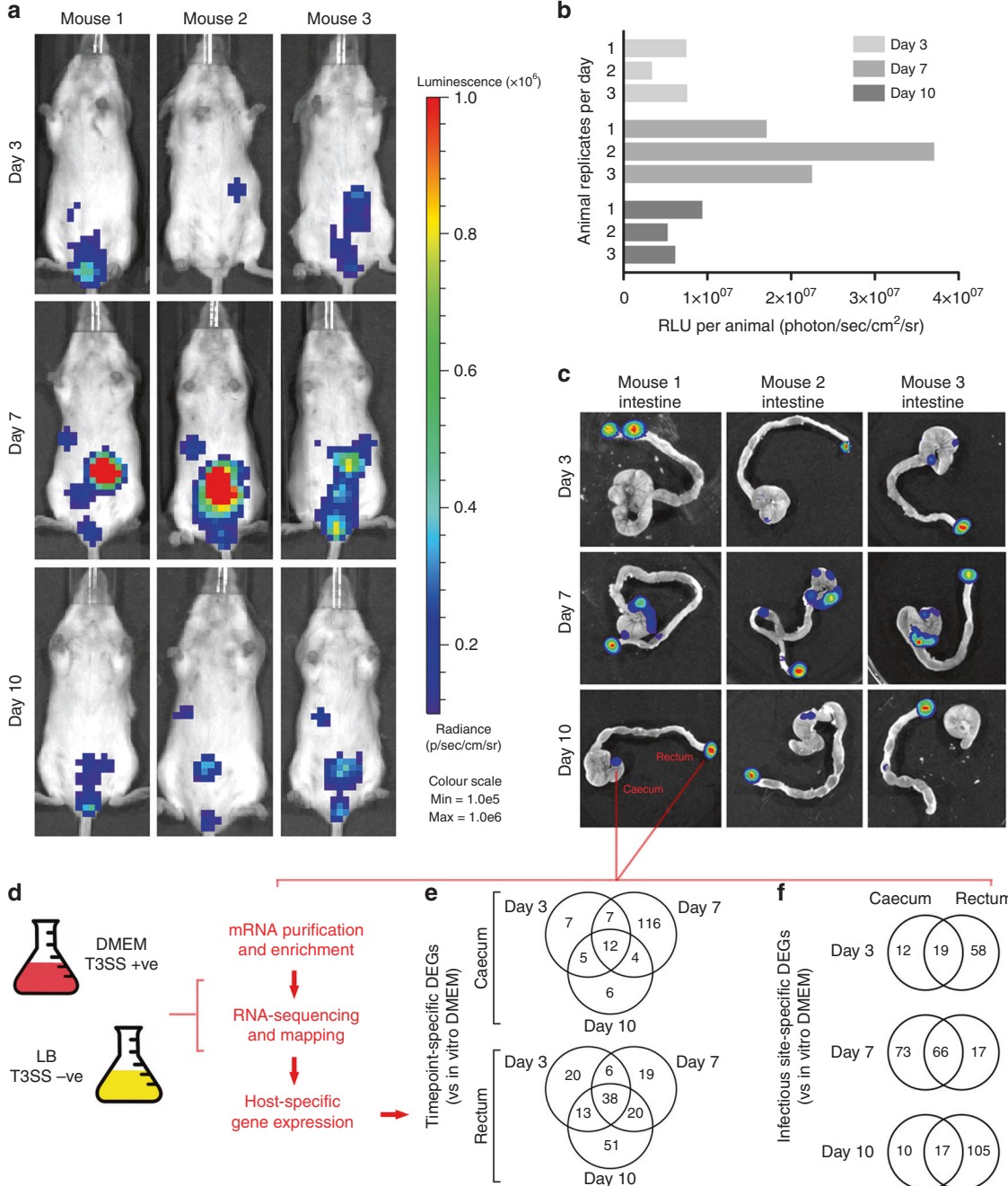

**Fig. 1** In vivo RNA-seq analysis of *C. rodentium* gene expression during murine infection. **a** Monitoring of murine colonization by *lux*-marked *C. rodentium* using IVIS. The scale of luminescence radiance (p/s/cm/sr) is normalized to a maximum of 1 × 10⁶. **b** Quantification of IVIS data for each animal infected as per panel **a**. **c** IVIS analysis of dissected intestines corresponding to animals from panel **a**, which were used to process localized bacterial mRNA from the caecum and rectum sites (highlighted in red). **d** Experimental workflow for in vivo RNA-seq analysis. Control conditions (DMEM/T3SS+ve and LB/T3SS −ve) were used as a comparator to infection derived transcripts obtained for *C. rodentium*. **e** Venn diagrams depicting the number of differentially expressed genes (DEGs; >1.5-fold change in expression versus growth in DMEM; $P \leq 0.05$ EDGE test) identified over the course of infection according to site of colonization. **f** Venn diagrams depicting the number of DEGs identified between sites at each time point

139 DEGs at peak infection (85 up; 54 down) and 27 DEGs at the late stages of colonization (12 up; 15 down). Conversely, at the rectum the number of DEGs identified increased over the course of infection with 77 DEGs identified at early (43 up; 34 down), 83 DEGs identified at the peak of infection (54 up; 29 down) and 122 DEGs identified during late colonization (75 up; 47 down).

One key question concerning bacterial pathogenesis is what drives the niche-specificity of host colonization? A/E pathogens are known to colonize unique parts of the gut, which exposes them to different environmental cues. EPEC colonizes the small intestine of humans, whereas EHEC colonizes the distal colon of humans and the terminal rectum of cattle. Accordingly, these sites contain different signals that modulate LEE expression[23]. To identify any site-specific transcriptional signatures, we compared RNA-seq profiles between the caecum and rectum. DEG overlap during infection revealed that most genes differentially expressed at each site were unique to the time during infection, with a small subset of DEGs present throughout infection (Fig. 1e). This

suggests that the pathogen may experience 'phases' of gene expression throughout its tenure within the host. Furthermore, a number of common overlaps were seen between the caecum and rectum in terms of differential expression compared to DMEM (Fig. 1f). These included genes from the LEE and certain non-LEE-encoded effectors (NLEs), which were differentially expressed at both sites suggesting a level of commonality in colonization mechanisms independent of subtleties in environmental composition (Supplementary Data 2).

**Expression dynamics of *C. rodentium* virulence factors in vivo.** The in vivo transcriptome of *C. rodentium* included expression of virulence determinants such as the LEE, *kfcC* (K99 fimbrial-like adhesin), *lifA* (lymphocyte inhibitory factor A) and the *croIR* quorum sensing system (Fig. 2a; Supplementary Data 1)[28–30]. Most strikingly was the strong upregulation of *kfcC* throughout infection, displaying upregulation of >700-fold at peak colonization (Supplementary Figure 3a/b). KfcC was previously found to contribute to the persistence of *C. rodentium* colonization in vivo and our data identified it as being the most upregulated ORF throughout infection. *kfcC* is directly regulated by the bicarbonate sensor RegA, which is constitutively active and expressed in vivo (Supplementary Data 1)[31]. Differential expression in vivo varied depending on site and phase of infection however expression of virulence factors dominated the transcriptome throughout (Supplementary Data 1). Indeed, profiling of the most abundant transcripts during infection revealed strong expression of LEE and NLE encoding genes, as well as housekeeping genes such as *lpp* (Brauns lipoprotein), *raiA* (translation inhibitor), cold-shock proteins and nucleoid-associated proteins H-NS and HU. These data suggest that regardless of the site, colonization is always dependent on the LEE-encoded T3SS and how this appendage is regulated in response to different environmental cues, owing to its known essentiality for colonization[18].

The LEE is organized largely into 5 polycistronic operons and in vitro can be induced entirely by growth in DMEM (Fig. 2b; Supplementary Data 3). However, expression dynamics of the LEE in vivo were more heterogeneous than that of DMEM. In general, patterns of expression between the caecum and the rectum were largely similar except for less significantly down-regulated LEE genes at the rectum, possibly reflecting preparation for the 'hyperinfectious' state of *C. rodentium* directly after shedding from the host (Supplementary Figure 3c)[32]. Expression of *LEE5* and *LEE4* particularly was enhanced in vivo. *LEE4* encodes the gatekeeper protein SepL, various chaperones and importantly the translocon proteins EspA, EspD and EspB (Fig. 2b)[14,33–36]. These ORFs showed particularly high expression at all stages in vivo. *espD* showed >8-fold higher expression at the infectious peak of the caecum (Fig. 2c). This was in contrast to the *LEE1*, *LEE2* and *LEE3* operons (exemplified by *escR*, *cesD* and *mpc* respectively) as well as non-operon encoded ORFs (such as *etgA*) that actually displayed decreased expression levels in vivo (Fig. 2c). Differential LEE expression was confirmed by qRT-PCR on tissue-derived RNA from peak infection (Supplementary Figure 3d). The *LEE1, LEE2* and *LEE3* operons largely encode structural components of the T3SS, whereas *LEE4* encodes the translocon and *LEE5* encodes Tir and Intimin[15]. The in vivo RNA-seq data suggests that, once formed, the T3SS structural genes are downregulated but the translocon and effector encoding genes are continuously expressed, likely reflecting niche maintenance after initial attachment. These data are important considering the recent discovery of bimodal expression patterns of EPEC LEE promoters in response to environmental stimuli in vitro as well as extensive post-transcriptional regulation that occurs at *LEE4/5*[37–39]. Taken together with our findings, this

suggests that the LEE is likely subject to epigenetic regulation in vivo, which results in fine-tuning of gene expression in response to the host environment.

**Host-signals co-ordinate expression of the effector repertoire.** The LEE is essential for colonization of the host (Supplementary Figure 1a) and it has recently been demonstrated that NLEs are essential for A/E lesion formation in vitro[18,40]. However, despite a great deal being known about LEE regulation, an understanding of the mechanisms and stimuli driving NLE regulation is lacking[23]. Using our RNA-seq approach, we defined the expression profiles of the NLE repertoire in vivo (Fig. 3a). Expression of distinct effector encoding genes was heterogeneous however we identified a subset of NLEs that displayed co-ordinated expression in vivo (Supplementary Figure 4a). Specifically, *espI* (also known as *nleA*), *espS*, *espO* and *espM3* were the most abundant NLE transcripts and had distinct expression patterns depending on the conditions (*espI*[nleA] and *espS* were induced in DMEM whereas *espO* and *espM3* were induced in LB), but were all significantly upregulated in vivo throughout infection, suggesting an element of co-regulation (Fig. 3b; Supplementary Figure 4b).

NLEs are often encoded on horizontally acquired genomic islands or carried by bacteriophage on exchangeable loci, accounting for the variable nature of the effector repertoire in A/E pathogens[19]. The genomic context of *espI*[nleA], *espO* and *espM3* is distinct (Supplementary Figure 4c). *espO* and *espM3* are encoded on the genomic islands GI11 and CRPr33, respectively. GI11 also houses the NLE genes *espT* and *nleG8*, whereas CRPr33 encodes a LifA homologue (*lifA2*). EspO and EspM are conserved in EPEC and EHEC whereas EspS is specific to *C. rodentium*[27]. The *espS* allele is located directly downstream of *espI*[nleA] in *C. rodentium* and is co-transcribed with the latter (data not shown). In order to examine the mechanisms governing distinct NLE regulation, we generated transcriptional reporters of *espO, espM3* and *espI*[nleA]*/S*, testing their expression in four defined LEE-regulator mutant strains (*Δler*, *Δhns*, *ΔregA* and *ΔgrlA*). Each reporter gave a condition-specific readout of transcription depending on the environment (Supplementary Figure 4d). Moreover, the transcriptional response to the regulator deletions was distinct in the case of each reporter therefore revealing that at the basic genetic level each NLE had unique regulatory mechanisms (Supplementary Figure 4e). Collectively these results revealed that irrespective of individual function or genetic context, the specificity of the host environment is responsible for the co-regulation of a suite of infection relevant NLEs.

Physiological roles for effector proteins are often elusive. In order to link infection-relevant transcriptional changes in NLE expression to a specific function we selected EspS for investigation. EspO has recently been shown to play a role in persistence by influencing the host immune response, while EspM orthologous have been shown to play a role in modulation of actin pedestals and tight junction localization[41,42]. No significant difference in murine colonization was observed between wild type *C. rodentium* and *ΔespS* mutant up to peak infection, with clearance of *ΔespS* faster than the wild type in the resolving phase but not reaching statistical significance (Supplementary Figure 4f). However, histopathological analysis of tissue from infected colon sections at peak infection revealed that *ΔespS* caused excessive elongation of epithelial crypts (Fig. 3c). A hallmark of *C. rodentium* infection is colonic crypt hyperplasia characterized by intestinal repair at the site of colonization and thus crypt elongation and mucosal thickening[3]. The average crypt length of uninfected mice was ~150 μm ($n = 184$) while infection with the wild type caused significant extension to that of ~200 μm

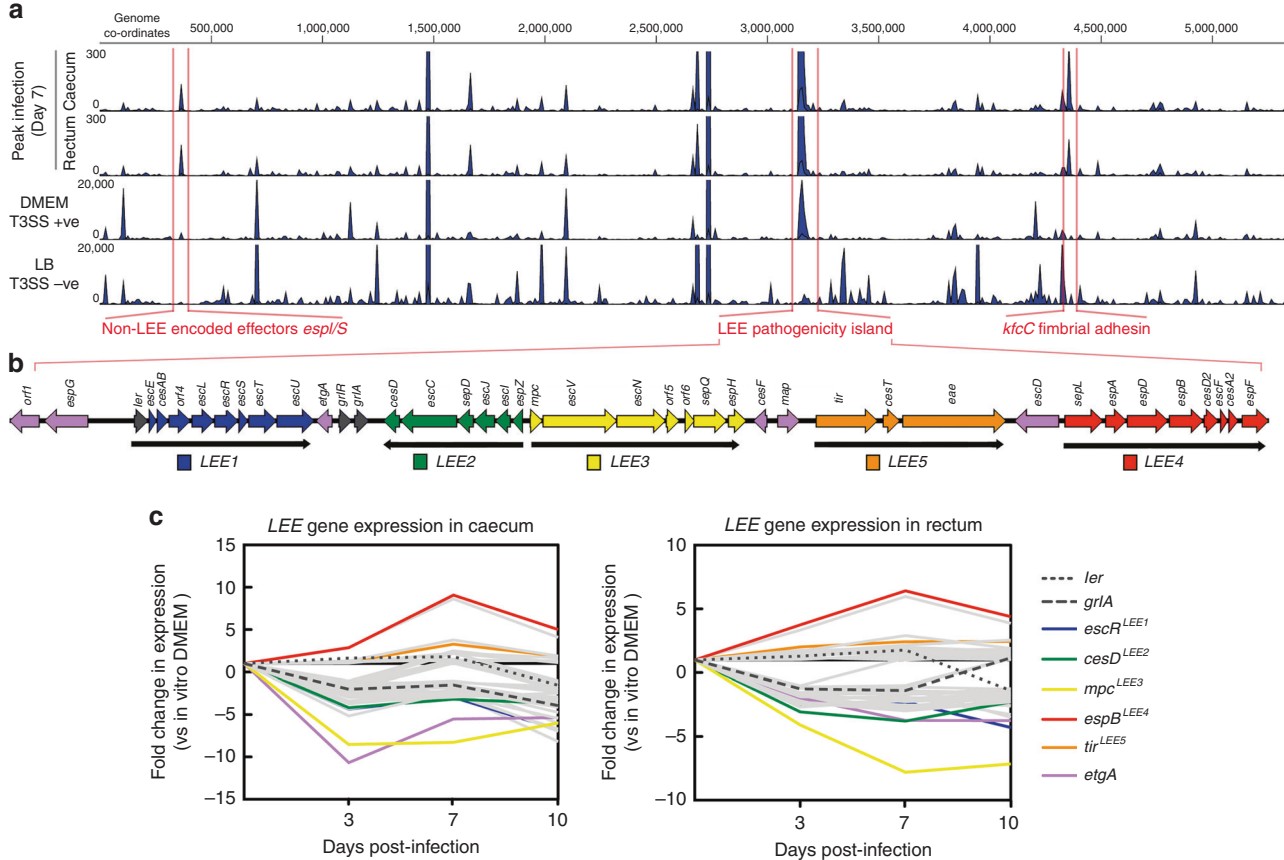

**Fig. 2** Dynamics of *C. rodentium* virulence gene expression in vivo. **a** Genome-wide coverage plots of RNA-seq data mapped uniquely to the *C. rodentium* reference indicating transcript levels in vivo. Data are representative of peak infection (day 7) and control conditions (T3SS+ve/−ve) with the relative scale indicated on the left. Regions of interest relating to virulence determinants are highlighted in red. **b** Expanded schematic illustration of the LEE pathogenicity island with each operon (*LEE1*–*LEE5*) colour coded. Regulator encoding genes (*ler* and *grlRA*) as well as LEE genes not encoded in operons are indicated in black or purple respectively. **c** Expression dynamics of the LEE in vivo compared to in vitro growth in DMEM. The line plots illustrate fold-change in expression at each site and time point during infection. Genes representing the highest fold change for each operon are colour coded according to panel 2b (and labelled in parentheses) while remaining LEE genes are illustrated in grey. Gene expression data represent the mean of three replicates in all cases and full expression heatmaps are found in Supplementary Figure 3c

($n = 382$), however excessive hyperplasia by Δ*espS* was characterized by significant crypt elongation to an average of ~230 μm ($n = 383$) ($P \leq 0.0001$, one-way ANOVA). To confirm that the alterations in crypt length were associated with increased colonocyte proliferation at the crypts we stained the tissue cross-sections with the cellular proliferation marker Ki-67 (Fig. 3d). Ki-67 scoring revealed that Δ*espS* displayed a significantly ($P \leq 0.0001$) increased expansion of proliferating colonocytes from the epithelial crypts, ~53% Ki-67 positive cells associated with wild type ($n = 155$) compared with ~63% associated with Δ*espS* ($n = 159$), thus accounting for the associated excessive hyperplasia and crypt elongation. We were unable to complement the Δ*espS* mutant in vivo, however the phenotype was consistent across two independently performed infections ($n = 10$). The induction of colonic crypt hyperplasia by *C. rodentium* drives luminal proliferation of the pathogen, which has recently been linked to oxygenation of the mucosal surface by proliferation of undifferentiated Ki-67 positive epithelial cells[43]. Our discovery that EspS may contribute to repressesion of hyperplasia reveals a novel function for an effector, playing a role in fine-tuning a phenotypic hallmark of *C. rodentium* infection. Therefore, our data suggest that co-ordinating gene expression of effectors with distinct roles is a key process in order to maintain physiological balance during infection.

**A *C. rodentium* metabolic signature induced during host infection**. The interplay between metabolism and virulence is critical for a pathogen to establish a niche within the host[1,2]. Having identified specific *C. rodentium* virulence factors co-ordinately expressed during infection, we next wished to uncover mechanisms governing their regulation. Gene ontology (GO) term grouping of DEGs revealed significantly ($P \leq 0.05$) enriched biological functions induced in vivo. At early and late infection, most enriched GO pathways identified were at the caecum (34 early and 33 late enriched terms), largely relating to processes such as amino acid biosynthesis, monosaccharide catabolism, metal ion homoeostasis, nutrient transport, respiration and stimulus response (Supplementary Figure 5a; Supplementary Data 4). A small number (12 early and 3 late enriched terms) of pathways were also enriched at the rectum including down-regulation of cofactor biosynthesis and cellular catabolism. During the peak of infection however, a greater number of significantly enriched functions were identified at the rectal site of colonization. DEGs that related to enriched GO terms at both sites were involved in processes such as nitrogen and monosaccharide metabolism, energy generation, amino acid biosynthesis and metal ion homoeostasis, reflecting the nature of the in vivo environment for survival and scavenging (Fig. 4a). Interestingly, rectum-specific GO terms during peak infection

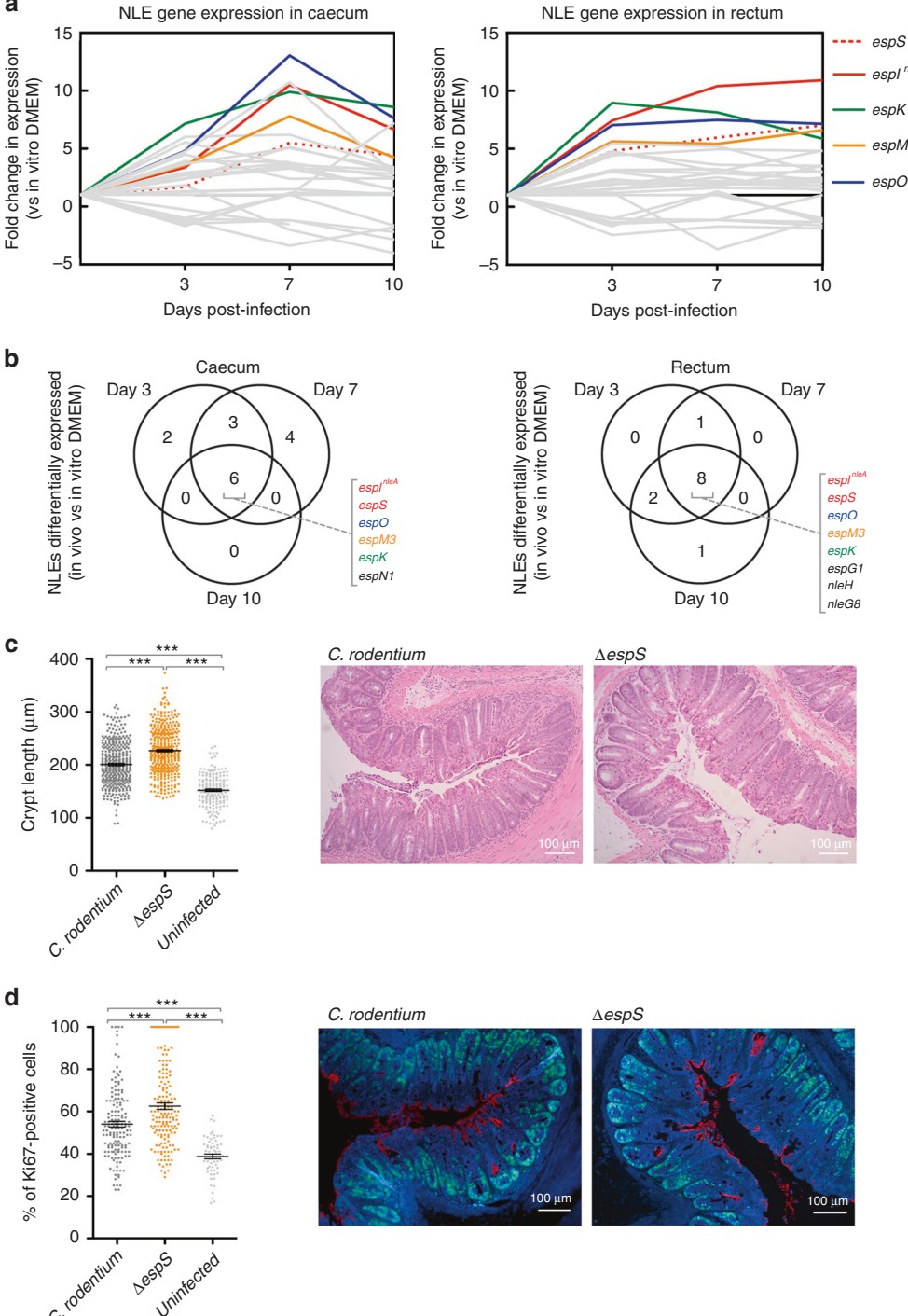

**Fig. 3** Co-ordinated expression of the non-LEE-encoded effector repertoire in vivo. **a** Expression dynamics of the NLE repertoire in vivo compared to in vitro growth in DMEM. The line plots illustrate fold-change in expression at each site and time point during infection. A selection of NLEs (*espS, espI^nleA, espO, espM3* and *espK*) displaying the highest fold changes in expression are labelled and colour coded while remaining NLE genes are illustrated in grey. Gene expression data represents the mean of three replicates in all cases and full expression heatmaps are found in Supplementary Figure 4a. **b** Venn diagrams illustrating the overlap in NLEs differentially expressed throughout infection ($P ≤ 0.05$; EDGE test). NLEs that were significantly differentially expressed in all in vivo samples are colour coded as in panel 3a. **c** Enumeration of colonic epithelial cell crypt length (μm) from mice (groups of $n = 10$; ± SEM) infected with either wild type *C. rodentium*, Δ*espS* or uninfected controls. Sampling was carried out at peak infection (day eight PI). Representative images of H&E-stained colon sections from wild type or Δ*espS*-infected mice are presented additionally. **d** Percentage measurement of epithelial proliferation in response to wild type or Δ*espS* infection by Ki-67 staining (±SEM). Representative images of stained colon sections from wild type or Δ*espS*-infected mice are presented. Markers used stained for *C. rodentium* (anti-Int280β; Cy3 red), Ki-67 (alexa-488 green) and host cells (DAPI blue). For all quantification, *** denotes $P ≤ 0.0001$ (one-way ANOVA with Bonferroni multiple correction test). Error bars represent standard error of the mean

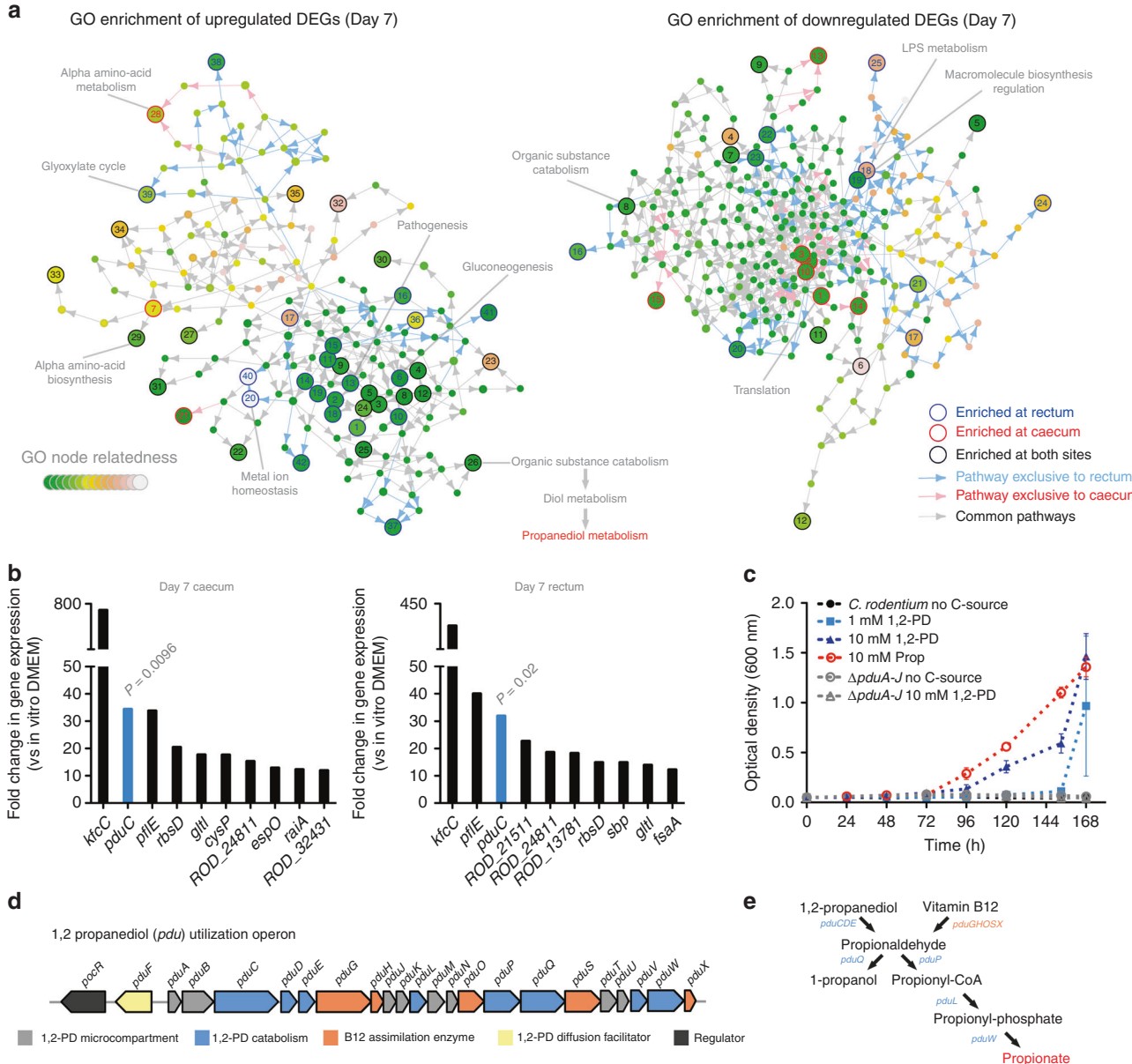

**Fig. 4** Host-associated *C. rodentium* metabolism co-ordinated with infection. **a** Networks graphs illustrating the identification of gene ontology (GO) terms in vivo relating to Supplementary Data 4. Networks were generated for both upregulated and downregulated genes individually. Coloured nodes represent GO terms associated with the DEGs in vivo and the arrows indicate relatedness to one another by distance. Significantly enriched GO terms ($P \leq 0.05$) were assigned a numbered node and each node is colour-coded to represent enrichment of a biological function at the caecum (red), the rectum (blue) or at both sites (black). The identification of 1,2-propanediol as a significantly enriched biological process in vivo is indicated at node 26. The colour coding of related nodes is indicated on the inset legend. **b** The top 10 DEGs identified at peak infection according to fold increase in expression (EDGE test). Genes from the *pdu* operon (1,2-propanediol metabolism) are highlighted in blue and the relative EDGE test *P*-value indicated above. Each bar represents the mean of 3 biological replicates. **c** Growth of *C. rodentium* and Δ*pduA-J* with 1,2-propanediol (1,2-PD) or propionate (Prop) as a sole carbon source. Data points indicate the mean optical density measurements (600 nm) of three biological replicates ± SEM. **d** The *C. rodentium pdu* operon. Genes are colour-coded according to role in 1,2-propanediol metabolism. The figure is adapted from Sturms et al.[80] **e** Schematic depiction of the 1,2-propanediol metabolic pathway. The intermediate products and genes involved in their generation are indicated

included gluconeogenesis and the glyoxylate cycle, the latter of which is essential for metabolizing fatty acids during gluconeogenesis[44]. Indeed, the LEE is promoted by aerobic and gluconeogenic conditions in response to the host-niche composition[45,46]. Regardless of site-specific subtleties in metabolism however, LEE and NLE expression was largely conserved between the two sites highlighting the niche-adaptable regulation of co-ordinated virulence factors.

Profiling of DEGs displaying the largest fold increases in vivo combined with GO filtering identified significant upregulation of genes belonging to the 1,2-propanediol (*pdu*) catabolic operon enriched at both sites of colonization (Fig. 4b; Supplementary Figure 5b). 1,2-propanediol is a product of fucose or rhamnose fermentation by the microbiota and can act as a carbon source promoting *Salmonella* invasion and proliferation during colitis[47,48]. We identified a significant increase in *pduB*, *pduC*

and *pduG* expression during infection (up to 70-fold) when compared with in vitro cultures, suggesting availability of this nutrient in vivo (Fig. 4b). When tested for the ability to grow on 1,2-propanediol as a sole carbon source, wild type *C. rodentium* grew on 1 mM with an increased growth rate at 10 mM (Fig. 4c). The *pdu* gene cluster is comprized of 21 genes co-transcribed polycistronically as well as a divergent regulator, and is carried by *C. rodentium* (Fig. 4d)[27,49,50]. A complete *pdu* operon is required to metabolize 1,2-propanediol to propionate, a short-chain fatty acid (SCFA) found in high mM concentrations in the intestine that we found can also support *C. rodentium* growth (Fig. 4e)[51,52]. As such, deletion of the genes *pduA-J* completely eliminated the ability to grow on 1,2-propanediol (Fig. 4c). These results identified 1,2-propanediol metabolism as an infection-relevant metabolic signature promoting *C. rodentium* fitness.

**C. rodentium virulence is regulated via 1,2-propanediol metabolism.** A/E pathogens utilize a myriad of signals present in the environment to modulate expression of the LEE and establish a host-niche[23,24]. We hypothesized that due to the induction of 1,2-propanediol metabolism in vivo, it may be sensed as a signal to modulate virulence. Growth of *C. rodentium* in DMEM supplemented with 1,2-propanediol had no significant effect on growth rate, allowing us to investigate effects on gene expression irrespective of a fitness advantage (Supplementary Figure 6a). Using a *C. rodentium LEE1* reporter (Cr*LEE1*:GFP), we found that *LEE1* expression was significantly enhanced ($P \leq 0.01$) in the presence of 1 mM 1,2-propanediol during exponential phase (Fig. 5a). A significant ($P \leq 0.05$) response could be seen as low as 0.05 mM, recently determined as a physiologically relevant concentration in mice mono-associated with the commensal *Bacteroides thetaiotaomicron* (Supplementary Figure 6b)[47]. It should be noted that the precise concentration of 1,2-propanediol in mice with a complete microbiota remains to be determined. Expression of a non-virulence associated housekeeping gene was unaffected (Supplementary Figure 6c). Propionate, the end product of 1,2-propanediol metabolism, has previously been shown to increase LEE expression and virulence in EHEC[53]. As a control for *LEE1* modulation, we tested Cr*LEE1*:GFP expression in the presence of propionate. *LEE1* expression was enhanced >2-fold in the presence of 10 mM propionate ($P \leq 0.01$), suggesting that *C. rodentium* senses intestinal SCFAs similarly to EHEC (Fig. 5a). We reasoned that due its ability to modulate *LEE1* and given the induction of *pdu* expression in vivo, 1,2-propanediol may act as signal co-ordinating several virulence factors including NLEs. All three NLE reporters (*espI*[nleA]/S, *espO* and *espM3*) showed significantly enhanced activity in the presence of 1,2-propanediol in stationary phase, indicating differential temporal regulation between the LEE and associated NLEs in response to this microbiota-derived cue (Fig. 5b). This suggests that 1,2-propanediol has the ability to co-ordinate the expression of distinct *C. rodentium* virulence factors.

A/E pathogens can respond to nutrient signals in the environment without the need to metabolize them[23]. We next investigated whether 1,2-propanediol acted as a regulatory signal directly. Analysis of *LEE1* regulation in the Δ*pduA-J* background demonstrated no enhanced expression in response to 1,2-propanediol, however enhanced *LEE1* expression of the wild type was restored either by complementation of *pduA-J* or by the addition of exogenous propionate (Fig. 5c). This suggested that metabolism of 1,2-propanediol to propionate was required to regulate virulence.

To assess the impact of 1,2-propanediol during clonization, we analysed attachment to HeLa cells with *C. rodentium* in the presence this signal (Fig. 5d). A significant increase ($P \leq 0.001$) in

the number of A/E lesions per cell was observed with an increase from an average of 16 lesions to 28 with 1,2-propanediol ($n = 220$ and 176, respectively). As a control, we also demonstrated a significant increase in A/E lesions in the presence of propionate ($n = 173$). Enumeration of A/E lesion formation by Δ*pduA-J* was similar to the wild type and indistinguishable in the presence or absence of 1,2-propanediol (Fig. 5e). However, addition of propionate resulted in enhanced colonization, with an average of 30 A/E lesions per infected cell ($P \leq 0.001$). The human A/E pathogen EPEC also carries a truncated *pdu* metabolic operon whereas EHEC does not. EPEC was unable to use 1,2-propanediol as a carbon source for growth, as was the same for EHEC (Supplementary Figure 6d). Accordingly, 1,2-propanediol had no effect on *LEE1* expression in EPEC or EHEC but propionate did (Supplementary Figure 6e). Collectively, these results suggest that the response to 1,2-propanediol is dependent on its metabolism and may indicate a host-specific metabolic adaptation for *C. rodentium* virulence.

**1,2-propanediol regulated virulence maximizes fitness in vivo.** To investigate whether 1,2-propanediol metabolism is essential in vivo we infected mice with wild type *C. rodentium* or Δ*pduA-J*. Colonization of the host by both strains was indistinguishable in the early and peak phases of infection with no phenotypic differences observed in tissue pathology (Fig. 6a; Supplementary Figure 6f). However, when colonization was further monitored, Δ*pduA-J* displayed a rapid clearance from the host in the resolving phase of infection whereas persistence of wild type *C. rodentium* was significantly higher ($P < 0.05$ at days 10–16), suggesting metabolism of 1,2-propanediol plays a key role during infection of the host.

In order to determine whether this fitness defect of Δ*pduA-J* was a result of the inability to use 1,2-propanediol as an energy source or a signal for T3SS regulation, we compared colonization dynamics of mice infected with *C. rodentium*[Pler-const] (a strain constitutively expressing the T3SS through a single base pair deletion in the −30 residue of the *ler* promoter) and a Δ*pduA-J*[Pler-const] derivative[54]. Colonization of mice ($n = 10$) by *C. rodentium*[Pler-const] followed comparable dynamics as the WT for the duration of the infection. Strikingly, Δ*pduA-J*[Pler-const] colonization was indistinguishable from that of the WT, suggesting that constitutive expression of *ler* (and by extension the T3SS) overcomes the loss of regulatory signals that occur in the Δ*pduA-J* background thus eliminating any fitness defect (Fig. 6b). Collectively, these data demonstrate clearly that endogenous 1,2-propanediol metabolism represents a pathogen-specific adaptation to the metabolic status of the host-niche primarily by fine-tuning virulence regulation leading to full persistence (Fig. 6c).

## Discussion

Overcoming innate host defences and the barrier of the resident microbiota in order to compete and establish a desired niche is at the forefront of an invading pathogens agenda during infection. Adapting to the metabolic status of the host and integrating this information with the regulation of virulence gene expression facilitates this process, thereby mediating niche recognition. We have probed the transcriptome of an A/E pathogen during infection of its natural host and discovered a novel regulatory mechanism used to fine-tune virulence in response to a host signal. *C. rodentium* responds to the microbiota-derived fermentation product 1,2-propanediol, activating the pathway responsible for its metabolism and subsequently uses this information to modulate virulence. This finding represents an

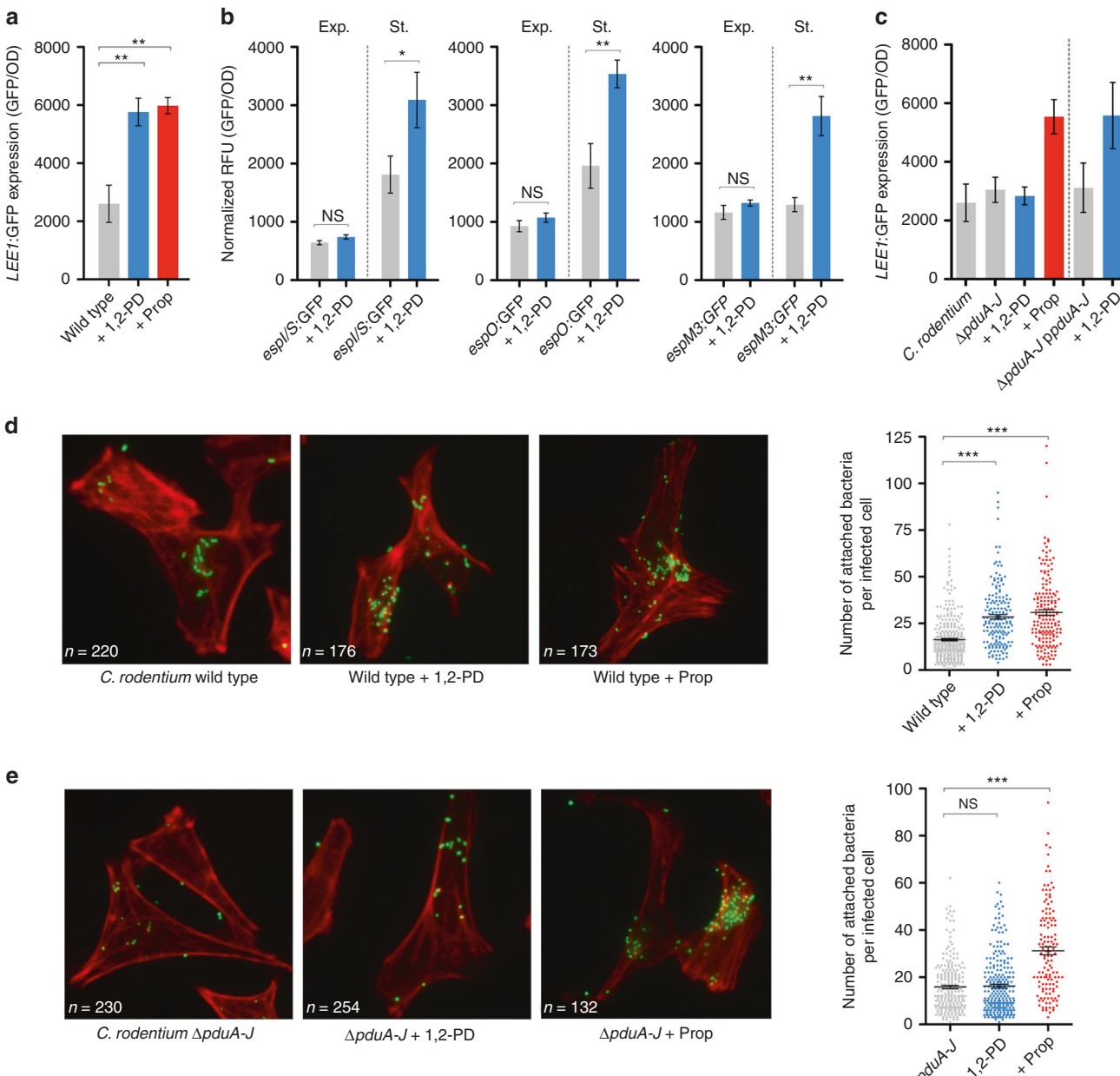

**Fig. 5** 1,2-propanediol metabolism fine-tunes *C. rodentium* virulence. **a** *LEE1*:GFP reporter activity in DMEM supplemented with 1 mM 1,2-PD or 10 mM Prop. Bacteria were sampled during exponential phase. *LEE1* expression was measured as GFP/$OD_{600}$. Data represents the mean of three biological replicates ± SEM. ** denotes $P \leq 0.01$ (Student's *t*-test). **b** Reporter assays using p*espI/S*:GFP, p*espO*:GFP or p*espM3*:GFP in DMEM with and without 1,2-PD supplementation. Data were collected at either exponential (Exp.) or stationary (St.) phase. Relative fluorescence units (RFU) were measured as GFP/$OD_{600}$ and the data represent the mean of three biological replicates ± SEM. *, ** and NS denote $P \leq 0.05$, $P \leq 0.01$ and not significant, respectively (Student's *t*-test). **c** *LEE1*:GFP reporter activity of Δ*pduA-J* in response to 1,2-PD or Prop. *LEE1*:GFP assays performed with Δ*pduA-J* carrying pBAD:*pduA-J* for complementation are separated by the grey dotted line. Data represent the mean of three biological replicates ± SEM. **d** Wide-field fluorescence microscopy images of attaching and effacing lesion formation on HeLa cells by *C. rodentium*, with and without 1,2-PD or Prop supplementation. Host cell actin is stained in red (phalloidin-568), whereas bacteria are green (transformed with p*rpsM*:GFP). Condensed actin can be seen around the area of colonization. The number of cells analysed from three independent experiments is indicated on each image. Enumeration of A/E lesions per infected host cell (±SEM) is indicated on the right. *** denotes $P \leq 0.001$ compared to the wild type (Student's *t*-test). **e** Analysis of A/E lesion formation on HeLa cells by the *C. rodentium* Δ*pduA-J* mutant, with and without 1,2-PD or Prop supplementation. Data were analysed as per panel **d**

important adaptation to a host-specific signal used as an energy source and subsequently as a cue to trigger virulence.

Competition for limited resources is a hallmark of infection. Indeed, an ecological niche within the host can be defined by its nutritional composition[55]. As such, *E. coli* pathotypes occupy distinct niches within the gut avoiding excessive competition with others having similar metabolic requirements[55,56]. It has recently been proposed that *C. rodentium* subverts host cell physiology and metabolism to create an aerobic microenvironment at the epithelial surface in a T3SS-dependent manner to promote expansion within the host[43,57]. Indeed, we identified transcription of *sucA* (α-ketoglutarate dehydrogenase) in vivo, an enzyme required for aerobic completion of the TCA cycle and previously described to be highly expressed during *C. rodentium* infection[43].

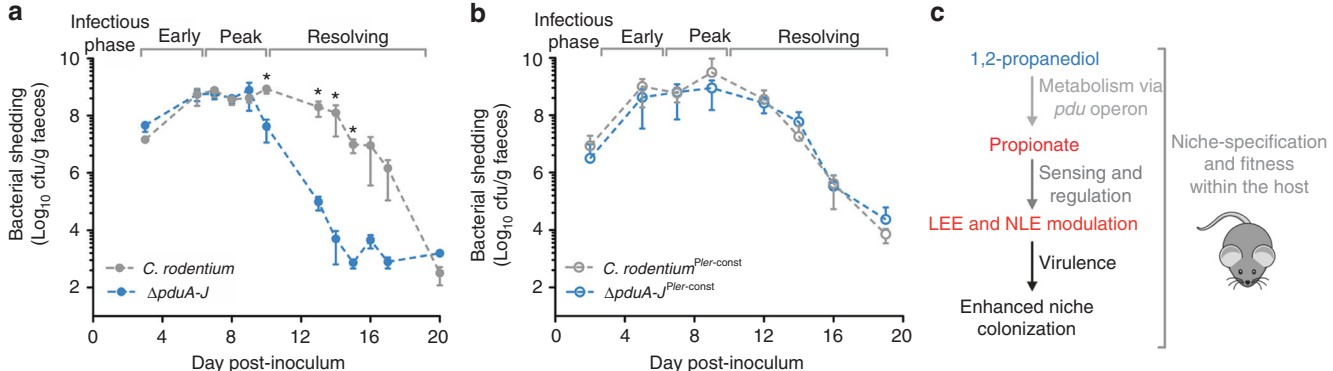

**Fig. 6** 1,2-propanediol regulated virulence is required for host colonization. **a** Colonization of mice host by wild type *C. rodentium* or Δ*pduA-J* (groups of *n* = 5). Each data point represents the mean bacterial load (±SEM) in faeces of infected mice (CFU/g). The infectious phase of each sampling point is indicated above the graph. Statistical significance between the strains at certain time points is indicated by a * denoting *P* ≤ 0.05 (Mann–Whitney *U*-test). **b** Colonization of mice host by *C. rodentium*[Pler-const] or Δ*pduA-J*[Pler-const] (*n* = 10). Each data point represents the mean bacterial load in faeces of infected mice (CFU/g). **c** Schematic summary of how 1,2-propanediol metabolism regulates virulence

More recently it has been demonstrated that *Salmonella* Typhimurium, which also takes advantage of an inflammation-induced aerobic gut-microenvironment, is capable of completing the TCA cycle in vivo through expression of succinate dehydrogenase (*sdhAB*) in the presence of an electron acceptor such as oxygen and consequently metabolizes microbiota derived succinate[58,59]. This is important considering both oxygen and succinate have recently been found to enhance LEE expression[46,60]. Our data identified significant upregulation of *sdhA* in vivo (up to 11.6-fold) providing further evidence that *C. rodentium* colonization promotes aerobic metabolism in the gut. This notion creates an added level of complexity that not only takes into account nutrient availability but also the aerobic capacity of the incoming pathogen.

Gene expression patterns can provide clues to pathogen behaviour in a given environment. Our approach led to the discovery of a metabolic pathway induced within the host that is required for colonization. Turner et al. previously applied a similar approach combined with large-scale transposon mutagenesis studies in a *Pseudomonas aeruginosa* murine burn wound model. Their study concluded that differential expression of bacterial genes within the host was only strongly correlated with an essential role in pathogen fitness for metabolic genes[61]. The interplay between metabolism and virulence is essential for A/E pathogens to colonize a host. For example, EHEC senses fucose through the FusKR two-component system and co-ordinately represses both LEE expression and fucose metabolism prior to appropriate niche recognition[62]. EHEC and *Salmonella* both utilize the nitrogen source ethanolamine as a direct signal modulating T3SS expression independently of metabolism[63–65]. Moreover, this ability to sense nutrients dictates spatial niche recognition in such pathogens. For instance, the high biotin status of the small intestine restricts LEE expression in EHEC but not EPEC[66]. EHEC also represses the LEE in response to D-serine, a carbon-source it cannot metabolize, in order to signal the presence of an unfavourable environment[67,68]. Furthermore, *C. rodentium* specifically responds to intestinal bicarbonate regulating the LEE in a pathotype specific manner[69].

The response to 1,2-propanediol represents a unique slant on this concept. Rather than being sensed as a direct signal itself, *C. rodentium* triggers 1,2-propanediol metabolism to the SCFA propionate. SCFAs are commonly produced in the host as end products of fermentation by members of the microbiota and act as diverse regulators of host physiology[70]. Furthermore, SCFAs can exhibit toxic effects on bacterial species at high concentrations leading to their use in food preservation. On the contrary, SCFAs can be used as sources of energy and signals for regulation of virulence. EHEC upregulates the LEE in response to intestinal SCFAs enhancing A/E lesion formation on host cells[53]. We have shown that *C. rodentium* enhances LEE expression in the presence of exogenous propionate but also that this regulation occurs through 1,2-propanediol metabolism, which results in generating and responding to endogenously derived propionate. Importantly, we found that the advantage of this strategy was entirely dependent on the ability to optimally express the T3SS in response to 1,2-propanediol metabolism rather by using it as a source of energy. Furthermore, this mechanism appeared to be crucial only at the latter stages of host colonization suggesting a temporal hierarchy in sensing of the niche. This implies that other environmental inputs into the LEE regulatory network determine its expression during initial colonization, whereas 1,2-propanediol metabolism becomes critical only during peak expansion of the *C. rodentium* population within the host. This suggests an intuitive mechanism whereby the pathogen generates its own positive cues for virulence regulation depending on the dynamic metabolic status of the host.

The mechanisms governing NLE regulation are not well understood. A/E pathogens are predicted to encode between 30 and 40 effectors translocated by the T3SS[19–21]. A small number of studies have investigated regulation of select NLEs. *espI*[nleA] encodes an essential effector and is thus co-regulated with the LEE. *espI*[nleA] is under direct control of Ler and H-NS. However, other effector-encoding genes (such as *nleB* and *nleC*) are under the control of a more cryptic regulatory mechanism[71]. Our approach defined the transcriptional status of the NLE repertoire in vivo, successfully identifying expression of effectors not previously identified using global proteomic approaches in vitro[20,21]. Importantly, we correlated the discovery of these strongly expressed effectors with a co-ordinated regulation in response to host-associated 1,2-propanediol. The molecular functions of effector proteins are often elusive, but we also discovered a novel role for a NLE (modulation of colonic hyperplasia), thus validating the relevance of our approach for the discovery of unknown virulence determinants.

This work presents a step forward in our understanding of how a bacterial pathogen behaves at the host interface. Pathogen transcriptome analysis during infection allowed us to discover a mechanism by which *C. rodentium* specifically fine tunes gene expression through metabolism, therefore regulating virulence in response to an intrinsic metabolite of its natural host. This study

has revealed further complexities of the infection process, offering novel insights into niche-adaptation of a bacterial pathogen.

## Methods

**Bacterial growth conditions**. Bacterial strains from pure stocks were cultured overnight in 5 ml LB media at 37 °C before being used to inoculate LB (T3SS –ve) or DMEM (Thermo Fisher) plus 1% LB and 5% $CO_2$ (T3SS+ve), culturing at 37 °C and measuring $OD_{600}$ for growth. For sole carbon source experiments, overnight cultures were first washed three times in PBS to remove trace LB and used to inoculate No-carbon E (NCE) media supplemented with trace metals (0.3 μM $CaCl_2$, 0.1 μM $ZnSO_4$, 0.045 μM $FeSO_4$, 0.2 μM $Na_2Se_2O_3$, 0.2 μM $Na_2MoO_4$, 2 μM $MnSO_4$, 0.1 μM $CuSO_4$, 3 μM $CoCl_2$, 0.1 μM $NiSO_4$) and either sodium propionate or 1,2-propanediol as a carbon source at the concentration indicated in the main text[51]. Experiments were performed at 37 °C with shaking and 200 nM cyanocobalamin was added with 1,2-propanediol during aerobic growth. No carbon source controls were used throughout to measure the background density of the inoculum. Antibiotics were used at the following concentrations: 100 μg/ml ampicillin, 50 μg/ml kanamycin, 15 μg/ml chloramphenicol, 75 μg/ml hygromycin, gentamicin 10 μg/ml and 500 μg/ml erythromycin. All chemicals were purchased from Sigma Aldrich.

**Bacterial strains, mutant generation and cloning**. The prototypical *C. rodentium* isolate ICC169 was used throughout this study and referred to as wild type[27]. EHEC TUV93-0 and EPEC E2348/69 strains were also used. Δ*pduA-J*, Δ*grlA*, Δ*regA* and Δ*hns* mutant derivatives containing non-polar gene deletions were generated using Lambda Red recombineering[72]. Briefly, the FRT-kanamycin cassette was amplified from pKD4 using primers containing 50 bp overhangs directly homologous to the up- and downstream regions of the gene of interest. The resultant PCR product was purified and DpnI treated. 1 μg of product was transformed into competent WT cells carrying pKD46 that were cultured in SOB media (100 μg/ml ampicillin; 30 °C) containing 10 mM arabinose to an $OD_{600}$ of 0.4. Subsequent recovery was carried out at 42 °C on LB-kanamycin (40 μg/ml) to eliminate pKD46 and select for successful recombinants. Positive mutants were identified by colony PCR and purified non-selectively at 37 °C. For Δ*ler* an adaptation of the Lambda Red method was used[73]. WT cells carrying pSIM18 cultured in SOB with arabinose (75 μg/ml hygromycin; 30 °C) were cultured as above before being heated to 42 °C for 15 min followed by rapid chilling in an ice water bath. The relevant PCR product was transformed into these cells by electroporation, recovered and positive knockouts selected as above.

The Δ*espS* mutant was generated by tri-parental conjugation. 300 bp homology flanks were cloned into pSEVA612S between I-*SceI* endonuclease sites to create pSEVA612SΔ*espS*[74]. Wild type *C. rodentium* were transformed with pACBSR encoding *SCEI* to create the receiver[75]. Donor strains were created by transforming CC118λpir with pSEVA612SΔ*espS*[76]. 20 μl of both donor and helper (*E. coli* 1047 pRK2013) strains were incubated at 37 °C on LB agar[77]. After 2 h, 40 μl of the receiver strain was spotted atop the donor/helper mix and incubated for a further 4 h. *C. rodentium* conjugants with pSEVA612SΔ*espS* integrated into the chromosome were selected by plating on LB with both chloramphenicol and gentamicin. Successful conjugants were cultured in LB with chloramphenicol and 0.4% arabinose for eight hours to induce I-*SceI* and excize the pSEVA612S backbone. Bacteria were streaked and selected for chloramphenicol resistance. Deletion of *espS* was confirmed by PCR. Deletion mutants were passaged several times in LB to cure the strain of pACBSR.

The complementation plasmid pBAD:*pduA-J* was constructed using the NEBuilder Hifi assembly protocol (New England Biolabs). Briefly, pBAD18 was linearized by PCR and two fragments corresponding to both halves of the *pduA-J* locus were also amplified by PCR using primers that created unique orientation-specific overhangs for assembly of the construct on the backbone of pBAD18. PCR fragments were purified by gel extraction and used in a one-step assembly reaction according to the manufacturers protocol. Reactions were transformed and selected on LB agar containing 100 μg/ml ampicillin. Positive clones were confirmed by digestion of the 7.4 kb *pduA-J* insert with *EcoRI* and *NheI*. All other complementation plasmids were constructed by cloning the gene of interest into pWSK29 (ampicillin) using ligation at the *SacI/XbaI* and *BamHI* restriction sites. A list of all strains, plasmids and primers can be found in Supplementary Tables 1-3.

***lux* marking of *C. rodentium* strains for in vivo imaging**. In order to visualize *C. rodentium* colonization during infection of mice, bioluminescent strains were generated using the p16S*lux* system. This method integrates the temperature sensitive p16S*lux* plasmid (containing the *luxABCDE* operon of *Photorhabdus luminescens*) into the target chromosome at the 16S locus[78]. Briefly, strains carrying p16S*lux* were cultured at 30 °C in LB (500 μg/ml erythromycin) overnight before subculturing 1:1000 into fresh media at 42 °C overnight. 100 μl of this was plated onto LB agar (plus erythromycin) and incubated at 42 °C. The temperature shift forces a low frequency recombination event and stable recombinants of the system were confirmed by subculture at 42 °C and imaging using the in vivo imaging system (IVIS) (PerkinElmer).

**Analysis of bacterial protein secretion**. Growth of *C. rodentium* in DMEM (plus 1% LB and 5% $CO_2$) at 37 °C induces expression of the LEE-encoded T3SS. To assay protein secretion into the surrounding medium 50 ml of cell culture supernatant at an $OD_{600}$ of ~0.7 was separated from the cellular fraction by centrifugation. Total secreted protein was precipitated from the supernatant with 10% trichloroacetic acid at 4 °C overnight. Secreted proteins were concentrated by centrifugation for 1 h at 4000 × *g*. The supernatant was removed and pellets resuspended in 50 μl of Tris-HCl (pH 8.8). Samples were normalized by $OD_{600}$ at the point of harvest and analysed by SDS-PAGE. The Δ*ler* mutant was used as a negative control for type 3 secretion. Bands corresponding to type 3 secreted proteins were confirmed by MS-MS analysis.

**Ethics statement**. Animal experiments were performed in strict accordance with the recommendations in the UK Home Office Animals Scientific Procedures Act of 1986 under personal project licence numbers 60/8797 and 70-8713. The experiments were subject to local ethical approval. All experiments were subject to the refine, reduce and replace consideration and all efforts were made to minimize suffering.

**Oral challenge of mice and bioluminescent imaging of infection**. Strains of interest were cultured in DMEM until an $OD_{600}$ of ~0.7 before being centrifuged and resuspended at 100× concentration in PBS. Groups of five BALB/c or C57BL/6 mice were then inoculated by oral gavage with 200 μl of PBS suspension ($3 × 10^9$ CFU confirmed by retrospective plating). For analysis of bacterial colonization abundance, stool samples were recovered aseptically and homogenized in PBS before serial dilution. The number of viable CFU per gram of stool was determined by plating onto LB agar with the appropriate antibiotic selection[10]. Infections were typically performed on two independent occasions and statistical analysis of CFU counts between groups was performed using the Mann–Whitney *U*-test. For whole-animal or organ bioluminescence imaging the IVIS SpectrumCT (PerkinElmer) was used. Regions of interest were identified and the total photon flux was quantified as photons/sec using Living Image software 4.3.1, package 1 (PerkinElmer).

**Staining and analysis of distal colon sections**. Tissue analysis was carried out as previously described[10]. 0.5 cm of distal murine colon was fixed in 10% formalin prior to embedding in paraffin, sectioning and mounting on glass coverslips. Sections were unwaxed in Histoclear (National Diagnostics, GA, USA), and rehydrated with sequential washes of 100, 95 and 85% ethanol, and PBS with 0.1% Tween-20 and 0.1% saponin (PBS-TS). Sections were boiled in 11.5 mM trisodium citrate with 0.05% Tween-20 for 20 min, blocked in 10% normal donkey serum in PBS-TS (PBS-TS-NDS) for 20 min and incubated with for 1 h with primary antibody solution: rabbit anti-Ki67 (PA5-19462, ThermoFisher scientific, MA, USA) (1:50), chicken anti-Int280β (1:50) in PBS-TS-NDS. Slides were rinsed twice in PBS-TS for 10 min before a further incubation with secondary antibody solution: DAPI (1:1000), goat anti-chicken Cy3 (Jackson Immunoresearch, UK) (1:100), donkey anti-rabbit Alexa488 (Jackson Immunoresearch) (1:100) in PBS-TS-NDS. Crypt length measurements were recorded using Zeiss ZEN blue software and the difference between groups was tested by one-way ANOVA with Bonferroni's multiple comparison test.

**In vitro HeLa cell adhesion assay**. HeLa cells were seeded on sterile coverslips ($10^4$ cells per coverslip in 24-well plates) in DMEM with 10% fetal calf serum and 1% Penicillin/Streptomycin. One hour prior to infection cells were washed three times with PBS and fresh DMEM was added with any supplementary antibiotics or additions described below. Cultures of wild type *C. rodentium* or Δ*pduA-J* (transformed with p*rpsM*:GFP) in DMEM alone and supplemented with either 10 mM sodium propionate or 1 mM 1,2-propanediol plus 200 nM cyanocobalamin were grown to an $OD_{600}$ of 0.7 before back diluting to 0.1 in DMEM. A volume of 10 μl of bacterial culture was added per coverslip. The plates were centrifuged at 125 × *g* for 3 min and incubated at 37 °C with 5% $CO_2$ for 6 h. The medium was replaced after 6 h to prevent excess acidification and infection was continued for a further 6 h. The cells were next washed five times and fixed by incubation in 4% paraformaldehyde for 15 min. The cells were permeabilized with 0.1% triton x-100 for 10 min before washing and staining for 20 min with phalloidin-Alexafluor 568. Coverslips were washed twice before mounting on Vectashield and analysing using a Zeiss Axioimager M1 and Zen Pro software. A/E lesions could be identified by condensation of host actin around the site of bacterial attachment. Host cell-associated bacteria were quantitated using the event counter tool in Zen and the total percentage of infected cells was also determined. Data were analysed by imaging 10 random fields of view from at least three coverslips.

**Isolation of bacterial RNA from tissue and mRNA enrichment**. IVIS data obtained from dissected intestines of infected mice revealed the major colonization sites of *C. rodentium* as being the caecal patch and the terminal rectum. In order to analyse the transcriptome at these sites 5 mm sections were dissected from each mouse intestine using a sterile scalpel. To maximize the integrity of the RNA, dissected tissue was immediately washed three times in sterile PBS to flush the luminal contents, immersed fully in RNA*later* (Ambion) and quickly separated according to

the caecum and rectum followed by incubation at 4 °C until extraction. As uninfected control mice did not have any measureable luminescence the equivalent regions were estimated and extracted in tandem. Extracted tissues were homogenized using a TissueLyser LT and 7 mm stainless steel beads (Qiagen) at 50 Hz for 3 min. Total RNA was extracted using the TRIzol Plus RNA purification kit (Ambion) according to the manufacturer's specifications and subsequently DNAse treated using DNAse TURBO (Ambion). In vitro control samples were cultured in triplicate to mimic T3SS positive (DMEM) or negative (LB) conditions. For these samples, 10 ml of bacterial culture was used to extract total RNA identically to in vivo samples. RNA concentration and integrity (RIN) were measured using a 2100 Bioanalyzer (Agilent) or LabChip GX (PerkinElmer). RIN scores of higher than 7 were considered excellent and obtained for all samples used in this study. In order to increase the subsequent resolution and provide a more accessible platform for sequence depth and multiplexing, aliquots of total RNA from infected tissues were subject to a two-step enrichment process prior to RNA-seq. First, the MICROBEnrich kit (Ambion) was used to deplete host polyadenylated mRNA and 18/26S rRNA thus enriching for bacterial total RNA. Second, host depleted samples were enriched for bacterial mRNA using the MICROBExpress kit (Ambion) in order to further deplete bacterial rRNA. This step was also performed for the in vitro control samples. All enrichment steps were carried out largely according to the manufacturer's specifications however due to enrichments from complex samples being an inefficient process, the volume of capture oligo and magnetic beads used was scaled up per reaction to help improve enrichments. The enrichment process was variable between samples and depletion of rRNA peaks was evaluated by Bioanalyzer/LabChip analysis.

**Illumina library preparation and RNA-sequencing.** Library preparation of RNA samples and sequencing was performed at the University of Glasgow Polyomics facility. Libraries were prepared using the TruSeq Stranded mRNA library prep kit (Illumina) according to the manufacturer's instructions. Sequencing was performed on a single run of the Illumina NextSeq 500 platform (75 bp length; paired-end).

**RNA-seq data mapping and differential expression analysis.** Quality control of RNA-seq reads was performed using FastQC (Babraham Bioinformatics) to assess sequence quality (minimum Phred threshold of 20) before being imported into CLC Genomics Workbench for processing and mapping. Reads were aligned to the *C. rodentium* chromosome (NC_013716.1) as well as associated pCROD1, pCROD2, pCROD3 and pCRP3 plasmid sequences (NC_013717.1, NC_013718.1, NC_013719.1, AF311902) downloaded from GenBank at NBCI[27], as well as *Mus Musculus* to seperate these reads from those mapping to *C. rodentium* and obtain data on the percentage of bacterial reads obtained. Read mapping was tested over a range of stringencies either side of default parameters a was finally performed with a mismatch score of 2, insertion/deletion cost of 3 and length/similarity fraction of 0.9 to improve specificity of mapping. To determine any bias of cross-mapping from highly conserved RNA sequences associated with residual members of the microbiota, we mapped enriched mRNA from both infected and uninfected mice to compare the mapping over the entire *C. rodentium* genome and annotated open reading frames (ORFs) only. This revealed a mapping fraction from uninfected mice that overlapped with *C. rodentium* un-annotated genome regions but very few reads mapping to *C. rodentium* unique ORFs (Supplementary Data 1). Examination of cross-mapping reads using BLAST revealed that they predominantly mapped to conserved tRNA or rRNA regions that are un-annotated or do not have a locus tag in the *C. rodentium* GenBank file. For this reason, reads mapped only to unique ORFs were considered for downstream analysis allowing transcriptome analysis with minimal bias from enrichment procedures or reads not unique to *C. rodentium*. Replicate correlation was measured by the Pearson and Spearman test in CLC and similarity in transcript abundance across genomic regions verified by genome-wide mapping coverage.

To identify differentially expressed genes (DEGs) between pairwise groups of replicates, we applied the exact test using the empirical analysis of DGE (EdgeR) tool implemented in CLC. This method normalizes the libraries and calculates differential expression based on raw sequence reads mapped. Genes were considered differentially expressed if they displayed an absolute positive or negative fold change of ≥1.5 and a Benjamini Hochberg-corrected *P*-value of ≤0.05 (5% false discovery rate). Because it could not be determined whether an ORF lacking or displaying only a small number of mapped reads was not expressed or simply a subject of sequence depth limitations, ORFs from in vivo transcriptomes with <5 reads mapped and in vitro samples with <20 reads mapped on average were not included in the analysis.

**Functional gene ontology (GO) analysis of RNA-seq data.** GO terms (biological process aspect) were assigned to the genes of the *C. rodentium* genome sequence by conversion of their Uniprot identifiers to GO terms using the EBI GOA database (https://www.ebi.ac.uk/GOA). For each condition in a comparison set, for 10,000 replicates *n* genes (the number of genes in the condition) were randomly sampled from the *C. rodentium* genes and the GO terms associated with these randomly selected genes were identified. Using the GO.db package within R we then identified the ancestors of each term in the biological process GO hierarchy and recorded these. These data provided a distribution of hierarchical GO terms that may be expected to be identified from a random selection of *C. rodentium* genes. For each of the actual genes identified per condition, we repeated this process

(extracting hierarchical GO terms) and derived *P* values by comparing how many times we identified a GO term in the actual genes with the number of times these GO terms would be expected to be encountered at random. These *P* values were adjusted for multiple tests via the Hochberg method.

Parent–child relationships in the GO hierarchy were converted to a graph format using the iGraph package within R. Large (numbered) vertices represent GO terms that are significantly over-represented in either group, red outline indicates over-representation in one group, blue in the other, and grey in both. Smaller nodes are GO terms that exists as ancestors of the significantly enriched terms, but are themselves significant. Red edges represent parent–child relationships that are only found in one group, blue edges represent parent–child relationships that are only found in the other, and grey edges represent relationships that are found in both. To provide a notional 'functional group' to each GO term, the GO definitions were mined using the tm package within R to reveal relationships between the words used to describe each of the GO terms. These relationships were estimated into a number of groups using k-means (1000 replications). The most abundant word found for each in the text for each group of GO term definitions was then used to colour the vertices and describe these groups.

**cDNA generation and qRT-PCR analysis.** In order to verify data obtained from in vivo RNA-seq quantitative reverse transcriptase PCR (qRT-PCR) was performed. Due to the limited amount of in vivo RNA available and the greater amount of template material required, a small but relevant selection of gene targets were chosen to verify the data. RNA obtained from the in vitro control conditions and RNA obtained from both the caecum and rectum of 3 mice at day 7 (peak of infection) was used for the analysis. RNA was normalized to ~100 ng and converted to cDNA using the SuperScript First Strand cDNA kit (Invitrogen) and qRT-PCR was performed using the GoTaq qPCR master mix kit (Promega) according to the manufacturer's specifications. qRT-PCR primer pairs were tested using serial dilutions of *C. rodentium* gDNA obtained using the PureLink genomic DNA kit (Thermo Fisher), with efficiencies being accepted at 95–105%. The housekeeping gene *gapA* was used as a calibrator for the analysis. qRT-PCR reactions were performed in technical and biological triplicate using the ECO real-time PCR system (Illumina) and data were analysed according to the $2^{-\Delta\Delta CT}$ method.

**Construction of GFP reporters and assay of promoter activity.** Regions containing ~300 bp upstream of the ATG as well as the first 6–8 codons were amplified by PCR using primers containing 5′ *Bam*HI and 3′ *Kpn*I restriction overhangs. These products were cloned into pAJR70 at a single cloning site to create in-frame fusions[79]. Reporter activity was measured by culturing the strain of interest carrying different reporter plasmids to a desired $OD_{600}$, then measuring both the cell density and absolute fluorescence output (excitation 485 nm; emission 550 nm). Readings were taken in black walled clear bottom plates using a FLUOstar Optima plate reader (BMG Labtech, UK). Data were corrected for background fluorescence by subtracting noise using bacteria carrying promoterless pAJR70. Promoter activity was determined as relative fluorescence units by dividing absolute fluorescence output by $OD_{600}$. Experiments were depicted as the mean value ± SEM and were compared to the equivalent value obtained for the WT for determination of statistical significance (Student's *t*-test). Experiments were performed in biological triplicate.

**Statistical analysis, software and data preparation.** RNA-seq data analysis was performed entirely using CLC Genomics Workbench version 7.5 and the FastQC application. Coverage graphs and Pearson Spearman plots were generated in CLC, and heatmaps were generated using Excel. GraphPad Prism version 5.0 was used to generate data charts and perform statistical analysis obtained from reporter assays and infection experiments. Cloning strategies were designed using MacVector version 12.5.

## Data availability

The raw data have been deposited to the European Nucleotide Archive under the accession numbers ERS1875884 to ERS1875911.

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

## Acknowledgements

We would like to thank Glasgow Polyomics for library generation and sequencing and Zoe Marjenberg for assistance with animal work. We are also very grateful to Professor's Jose Penades and Brett Finlay for helpful discussions regarding the manuscript. This work was supported by a Wellcome Trust Investigator award to Gad Frankel (107057/Z/15/Z), Wellcome Trust ISSF grants (105614/Z/14/Z and 204820/Z/16/Z) and BBSRC funding (BB/M029646/1 & BB/R006539/1) awarded to A.J.R.

## Author contributions

J.P.R.C., A.J.R. conceived and designed the research. J.P.R.C., S.L.S., N.O'.B., G.R.D., V.F.C. performed the experiments. J.P.R.C., S.L.S., R.J.G., G.R.D. analysed the data. D.R.G., P.H., D.G.E.S., G.F. contributed analysis tools and reagents. J.P.R.C., A.J.R. wrote the paper.

## Additional information

**Competing interests:** The authors declare no competing interests.

