## [Peer Review File · Nature Communications]

Reviewers' comments:

Reviewer #1 (Remarks to the Author):

This is a very nice and important study, where the authors investigate the transcriptional profile of *C. rodentium* during murine infection. They find, as predicted, upregulation of the virulence genes, and discover that effectors that are differentially regulated in vitro are coordinately regulated in vivo. They also find that the 1,2 propanediol metabolic pathway is upregulated during infection, and that 1,2 propanediol and the SCFA propionate up-regulate LEE gene expression. There is a lot of interesting and valuable information to the field in this manuscript, but some clarifications would improve it, as well as the addition of some controls and some experiments to solidify the proposed model.

Main points:

The main issue here would be to figure out whether the 1mM of 1,2 propanediol used to increase LEE gene expression in vitro is physiologically relevant. This should be measured in the cecum and the rectum.

Another main point is to address whether during murine infection 1,2 propanediol is mostly important as a metabolite or as a signal for virulence gene regulation. They run an experiment with a pdu mutant assessing colonization by CFUs on Fig. 6F, but how about some other markers of disease? How is survival? Colonic hyperplasia? Pathology? Virulence expression?

Moreover propionate is an SCFA derived from the microbiota. How does the pdu mutant fare compared to WT in germ free or microbiota depleted mice? Would this phenotype go away?

Minor points

Fig 6A needs a negative control, e.g. a gene that does not respond to 1,2 PD or Prop.

Figs 6B needs statistics in the stationary phase measurements. Are these changes significant?

Fig 4D: Other pathology parameters should also be scored.

Complement ler, regA, grIA and hns mutants for the in vitro measurements

How is the growth of the hns mutant? Is it impaired?

The use of only Esp nomenclature can cause confusion. I figured EspI is NleA, adding this in parenthesis would be nice. Specially because NleA is regulated by Ler, and it would make it easier to correlate with literature that refers to this gene as nleA. People who are interested in NleA would miss this. I assume EspS has been described for the first time here, please clarify. This is actually a pretty cool point, because Citro causes hyperplasia and EHEC and EPEC don't. If the other two Esps also have an nle name please add it. These things help to the correlation with genomics papers and the Tobe PNAS 2006 nle paper.

Reviewer #2 (Remarks to the Author):

In the present study, the authors established a method to globally profile gene expression of the murine pathogen *Citrobacter rodentium* during infection of an in-vivo mouse model. Particularly, they isolate total RNA from the caecum and rectum of infected mice, i.e. two preferred *Citrobacter*

There should be more information in the main text of the manuscript describing the sampling of the RNA, given that it represents the major dataset of the study. E.g. it is not mentioned in the main text how many replicates were taken and analyzed.

Lane 143 and following: How much % of these were of *Citrobacter* origin (what read number refers to 100% in Suppl. Fig. S2B)? Only mentioning that each library was sequenced to 26-45 Mio reads (lane 143) and that ~25% of the *Citrobacter* reads were uniquely mapped to ORFs (lane 146) is highly misleading as it implies a high *Citrobacter* sequence coverage, which is not the case as mentioned above.

Lane 160: It's true that *regA* mRNA was detected, but it was not induced relative to the in-vitro samples (unlike *kfcC*). What's known about *RegA*-mediated activation? E.g. is *RegA* activated at the protein level, or by phosphorylation? Please give more information here.

Lane 183: I am not sure if the abbreviation "NLE" is ever explained in the text.

Lane 213: This sentence suggests that there is qRT-PCR data for *escL* and *escV* in Fig. S3D, which I couldn't find (neither in Fig. S3D nor elsewhere).

Figure 2A occupies a lot of space but contains rather limited information. A simple line plot (or heatmap) showing expression of these genes would do. The read coverage plot might be moved to the supplementary material instead.

Figure 2B: This table can be moved to the supplementary material, as the numbers of DEGs may be inferred from panels C + D.

Figure 2C: Please check numbers again. E.g. caecum, day 3: 32 DEGs in panel B, 33 in panel C, 32 in panel D.

Figure 3: It is often unclear (and for the reader very confusing!) why sometimes RPKMO-values and sometimes fold-changes in gene expression are plotted. I'd strongly recommend to stick to one way of visualizing gene expression (preferentially, chose one of the 2 in-vitro conditions as a reference and display the EdgeR-calculated gene expression changes throughout the study).

Figure 3A: Are the data in the heatmap representative for 3 replicates, or does it display the mean expression from all 3 biol. replicates. This is an important difference! While it was fine to show a representative image of the read coverage in Fig. 2A, due to space limitations), there is no reason for why not to plot the mean from all 3 replicates in this heatmap (rather than 1 representative replicate). Please clarify/change accordingly (same goes for Fig. 4A).

Figure 3B: What does "undetected" in the x-axis label mean? Doesn't the materials & methods section say that these genes were filtered out?! In general, this panel is little informative and may be moved to the supplementary material.

Figure 3C: The legend says that LEE operons "that were differentially expressed" are shown. Differentially expressed compared to what? The legend further says that the genes *escL*, *escD* and *escV* are representative examples. Again: representative for what? Please, be more specific. Similar examples include the legend to Fig. 4B and Fig. 5B.

Figure 3D: In the opinion of this reviewer, this is not worth a dedicated model figure (or at least, it should be moved to the supplementary material).

Figure 4: For NLE genes for which the regulator was not identified, wouldn't the global, temporal expression data be predestined to perform a correlation analysis (of the respective effector with known members of global regulons) to predict those regulators?

Fig. 5B: This doesn't look like high fold-changes for many genes (close to 0, according to the color-code), even though according to the description the top-induced genes were plotted. Please clarify (and better explain in the legend or modify figure accordingly; same goes for Supplementary Figure S5B). In general (instead of panels B +C) it would be more intuitive to show 2 line plot diagrams (1 for caecum, 1 for rectum), with expression changes of all genes over the time-course as grey lines and highlight the pdu genes in colors. This way, the reader can much better put the expression of the respective genes into context).

Figure 5E: Please explain the abbreviations (e.g. "PD", "BMC").

Figure 6E: The WT strain should be included as a reference.

Figure 6F: It's stated nowhere how many replicates were analyzed.

Supplementary Figure S3A: Why not show most-INDUCED genes compared to in-vitro (rather than most ABUNDANT)?

Supplementary Figure S5A is little informative. Is it really needed?

niches in vivo, at three distinct time points, enrich for bacterial mRNA, convert them into cDNA libraries, and sequence them. This allows the authors to follow the expression of *Citrobacter* virulence genes, such as locus of enterocyte effacement (LEE) and non-LEE (NLE) genes, over time. Deletion of one previously largely uncharacterized NLE effector (EspS), that was highly expressed at all time points and at both tissues, led to increased crypt lengths in the infected intestine as compared to wild-type-infected mice, although the molecular mechanism remains elusive. Besides, the authors identify genes involved in *Citrobacter* 1,2-propanediol (1,2-PD) metabolism to be strongly expressed during the peak of infection and uncover a link between 1,2-PD and LEE expression. Interestingly, not 1,2-PD itself, but its conversion into propionate and sensing of the latter was responsible for LEE induction, and deletion of the pdu operon, encoding genes required for 1,2-PD to propionate conversion, facilitates *Citrobacter* clearance by the infected host.

Major comments:

Reviewing this manuscript, I was specifically asked to evaluate the RNA-seq-related parts of the study. Sampling, RNA isolation, bacterial mRNA enrichment (MicrobEnrich, MicrobExpress), cDNA library generation and the sequencing protocol are state of the art. However, I was surprised by the tiny fraction of reads that mapped to *Citrobacter* mRNAs (typically <0.1% of the total reads in the in-vivo samples). As the samples were sequenced to ~35 Mio total reads per library, the fraction of "useful" reads (mapping to *Citrobacter* ORFs) is in the range of 1,000-80,000 reads. This is extremely little! How do the authors explain that? Was the enrichment of bacterial mRNAs not efficient? What are the remaining 99.9% of the reads? Do they map against the murine genome, against *Citrobacter* sequences (but to multiple loci, and thus got discarded as the protocol selects for uniquely mapped reads), or against the genome of other (non-*Citrobacter*) bacterial species (members of the mouse microbiota)? In general, it is highly recommended that the authors include a separate supplementary table (or expand dataset S1) to include seq. depth and mapping stats for all samples (input reads, quality-filtered reads, mappable reads, uniquely mapped reads, etc. for all libraries).

Differential expression analysis is performed using EdgeR, which is state of the art (although I was surprised by the number of significantly differentially expressed genes called by EdgeR (around 100 per comparison), given the low sequence coverage of the in-vivo samples). In general, however, the (differential) expression data is presented in a rather confusing fashion: It is unclear to this reviewer why sometimes RPKMO-values are plotted and sometimes EdgeR-calculated fold-changes, and I fear that it will be even more confusing for readers less familiar with RNA-seq. Given that the authors anyways filter out lowly expressed/unexpressed genes and EdgeR considers abundance of a transcript for differential expression calling, why don't the authors just entirely rely on the EdgeR fold-changes and plot these throughout the manuscript?

The figures are dominated by heatmaps for visualizing expression data, which is not ideal. Since the authors have time-course data, I'd strongly recommend they select for one in-vitro reference control, and convert their heatmaps into line plots (reference → day 3 → day 7 → day 10; each one diagram for the respective tissue). Also, plotting the expression kinetics of all genes in light grey and highlighting the ones referred to in the main text in color on top, would allow the reader to put expression patterns of e.g. LEE and NLE genes, or the pdu operon, in context with the global expression changes during the time-course of infection.

Minor comments:

Wording: RNA-seq measures the steady-state levels of cellular RNAs, i.e. the sum of transcription AND decay. In contrast, the authors speak about "transcriptional landscape", "transcriptional activity" and similar throughout their manuscript (examples include, but are not restricted to, lanes 139, 415, 473, 483, 855, 870, 878, 881). This should be strictly avoided and replaced by terms such as "transcript levels", "expression" (or similar).

What makes it difficult to follow the flow of the story is that the main text sometimes appears to refer to the wrong figures/panels. Examples are lane 194 (Fig. S2B must be Fig. S3A), lane 212 (Fig. S3C must be Fig. 3C), lanes 865 + 867 (panels E and F must be panels C and D), lane 950 (6E is probably rather be 6C).

Reviewer #3 (Remarks to the Author):

In this work, the authors investigated host-derived signals that are used as cues by *Citrobacter rodentium* to help regulate bacterial gene expression required for adaptation to the host environment. They found that the metabolism of 1,2-PD derived from the microbiota was involved in a signaling/sensing system that coincided with the expression of the T3SS and associated effector proteins. The authors speculate that the metabolism of microbiota-derived 1,2-PD signals directly to *Citrobacter* in order to increase fitness in the host environment through the effects it has on bacterial T3SS gene expression.

Below are some areas that the authors may wish to consider in order to strengthen the findings and conclusions.

For the *in vivo* transcriptional profiling of CR, the method hinges on the ability to detect transcripts derived specifically from CR. However, given the high degree of orthologous genes between CR and other Enterobacteriaceae, how can the authors be sure that these RNA signals are derived from CR and not other members of the microbiota? While I understand that LEE genes would not be present in the normal microbiota, what does “high stringency” read mapping actually mean and how does it exclude non-CR RNAs from orthologous genes that are not unique to CR? This is particularly important in the later part of the manuscript when highly common genes involved in metabolic flux are introduced.

Figure 1 – the colonization dynamics of CR in resistant mice is well known, as is the location of colonization over time. This figure does not add anything to the literature that is not already known. I would be more interested (or concerned) with whether the constitutive expression of luciferase changes the tissue specificity, bacterial density, inflammation, or clearance rates of the CR compared to the uncontrived wild-type bacteria, as it is known that such reporters can alter fitness etc *in vivo*.

The idea that the time when bacteria are sampled *in vivo* would alter their transcriptional profile is not all that surprising. Would it not make sense to try and correlate these different gene expression signatures with the host response and/or bacterial density, which are both changing dramatically over 3-10 days post infection?

The narrative around the dynamics of LEE gene expression *in vivo* (Figure 3) – in part C, without error bars or a statistical analysis I’m not sure I would agree that *escL* and *escV* “decrease” over time; nor is the biological significance of this understood, as the bacteria are presumably replicating and being cleared over this time period in a highly dynamic fashion.

Without being able to complement your *espS* mutant regarding crypt length phenotypes, I would be cautious in saying that *EspS* represses hyperplasia *in vivo*. There could be numerous other explanations for this phenotype. It is not clear even what this line of investigation has to do with the question being addressed in the manuscript.

What is the 1,2-PD concentration *in vivo* in your experimental conditions? With 10 mM leading to a ~2-fold increase in gene expression *in vitro*, I’m concerned about the biological significance of this potential signaling pathway. Have the authors considered previously published work showing that propionate inhibits T3SS gene expression in *Salmonella* (Lawhon, 2002, *Mol Micro*), which is an opposite effect to that claimed here for CR?

The fact that a CR mutant unable to metabolize 1,2-PD had no colonization defect at the peak of

infection (when T3SS would be most needed and most active), seems to contradict your conclusion that this secondary metabolite is needed to activate T3SS. What happens to T3SS gene expression in vivo when you use this pduA-J mutant? According to your hypothesis, T3SS gene expression should be reduced, as would colonization.

We would like to thank the reviewers for both their positive remarks and detailed critical evaluations of the manuscript. We have taken the comments very seriously and attempted to address all concerns (as detailed below) by clarifying the points raised in our responses and, where appropriate, by adding extra experimental data or modifying the manuscript. We feel that the review process has resulted in a much-improved manuscript and that all weaknesses of the paper have now been clarified. We hope that this satisfies the reviewers' concerns and will be looked upon favourably.

Reviewer #1 (Remarks to the Author):

This is a very nice and important study, where the authors investigate the transcriptional profile of *C. rodentium* during murine infection. They find, as predicted, upregulation of the virulence genes, and discover that effectors that are differentially regulated *in vitro* are coordinately regulated *in vivo*. They also find that the 1,2 propanediol metabolic pathway is upregulated during infection, and that 1,2 propanediol and the SCFA propionate up-regulate LEE gene expression. There is a lot of interesting and valuable information to the field in this manuscript, but some clarifications would improve it, as well as the addition of some controls and some experiments to solidify the proposed model.

Many thanks to the reviewer for the positive comments.

Main points:

The main issue here would be to figure out whether the 1mM of 1,2 propanediol used to increase LEE gene expression *in vitro* is physiologically relevant. This should be measured in the cecum and the rectum.

This is an important point raised. When we began this line of enquiry there was no published information on 1,2-PD concentrations *in vivo*. As such, 1 mM was selected as a concentration to test for a phenotype *in vitro*. We reasoned that this was appropriate given the much higher concentrations of other metabolites (such as SCFAs) found in the gut and also given that *in vitro* culture is an unrealistic view of the complex environments that would actually be encountered *in vivo*. However, since then Faber *et al.* (2017. PLOS Path 13(1): e1006129) determined the concentration of 1,2-PD to be approximately 0.05 mM in the gut contents of mono-associated mice with *Bacteroides thetaiotaomicron* (a 1,2-PD producing strain). We therefore tested the response of the *LEE1* reporter in a range of 1,2-PD concentrations and found that we still obtained a significant response as low as 0.05 mM, making the finding physiologically relevant. This data has now been added to Figure S6. Furthermore, it should be noted and has been pointed out by others (Jakobson *et al.*, 2017. PLOS Comp Biol 13(5): e1005525) that the concentration determined by Faber *et al.* was determined from mono-associated mice and does not reflect the 1,2-PD concentrations that would be found, and likely higher, in mice harboring a full, natural microbiota.

Another main point is to address whether during murine infection 1,2 propanediol is mostly important as a metabolite or as a signal for virulence gene regulation. They run an experiment with a *pdu* mutant assessing colonization by CFUs on Fig. 6F, but how about some other markers of disease? How is survival? Colonic hyperplasia? Pathology? Virulence expression?

This is another valuable point. The prediction would be that, seeing as 1,2-PD (and propionate) can modulate LEE expression without enhancing growth rate *in vitro*, the effects on murine colonization are mediated through virulence regulation. However, it is possible that reduced fitness may account for some of the phenotype also. To address this, we have performed murine colonization experiments using strain ICC1370 (*C. rodentium* modified to constitutively express *ler*) and a *pdu* mutant derivative in order to determine if constitutively expressing *ler* can overcome the colonization defect observed in the WT *pdu* mutant. The new data (Figure 6) show that even when *pdu* is deleted the strain will colonise the same as the WT if *ler* is constitutively expressed. These data firmly support a model in which the major role of *pdu* is mediated through changes in the regulation of the T3SS.

Regarding the effects on other markers of disease, there was no significant difference in hyperplasia between the WT and *pdu* mutant as already indicated in Figure S6f. Survival was the same for all mice as the model is not lethal.

Moreover propionate is an SCFA derived from the microbiota. How does the *pdu* mutant fare compared to WT in germ free or microbiota depleted mice? Would this phenotype go away?

We reasoned that an experiment using *C. rodentium* in germ free or microbiota depleted mice would likely not address if any fitness defects were present in the *pdu* mutant *in vivo* (due to the lack of 1,2-PD) however a bigger issue is that colonization of germ free mice by *C. rodentium* is not dependent on the LEE and has vastly different dynamics to a natural infection (Kamada *et al.*, 2012. Science 8; 336(6086): 1325–1329). We therefore chose not to perform this experiment in order to avoid convoluting the results we have already described using mice with a natural microbiota.

Minor points

Fig 6A needs a negative control, e.g. a gene that does not respond to 1,2 PD or Prop.

We addressed this point with an *rpsM::GFP* reporter commonly used to assess housekeeping gene expression not related to virulence. 1,2-PD had no effect on expression of this gene. The data has been added to Figure S6.

Figs 6B needs statistics in the stationary phase measurements. Are these changes significant?

Reporter experiments were carried out in biological triplicate as detailed in the methods. This particular experiment did not obtain significant differences in each case, although the shifts in expression were consistently increased. This may be due to the NLE reporters producing weak signals or variation in stationary phase expression. To add robustness to the result we performed the experiment again over 3 more replicates and added this data. Additional replicates confirmed the results and added confidence to the data with statistical significance ($P < 0.05$). The figure and text associated has been modified to illustrate this.

Fig 4D: Other pathology parameters should also be scored.

The hyperplasia phenotype was validated by repetition of the infection experiment and by scoring both crypt length and ki-67 staining in all cases but the precise mechanisms governing the phenotype are beyond the scope of the study. Given the density of data in the manuscript and the fact that this experiment was not a main result and was included to exemplify that effectors expressed/regulated *in vivo* have physiological phenotypes, we have chosen to keep this section short and to the point. We would prefer to keep this section unmodified as a detailed evaluation of EspS would warrant an entire study on its own.

Complement *ler*, *regA*, *g1A* and *hns* mutants for the *in vitro* measurements

The effects of *ler*, *grlA* and *hns* on *nleA* regulation in EHEC/EPEC are known and have been complemented successfully (Garcia-Angulo *et al.*, 2012. J Bact 194(20) 5589-5603). The effects of these regulators on *espO* and *espM3* in the context of this study are largely independent of the *ler* regulon. This experiment was simply used to demonstrate different regulatory mechanisms for different effectors. Given that the result of the experiments makes this point as is, that this experiment was supplementary to the main text and that the *nleA* regulation has been complemented previously, we wish to leave the figure as is for simplicity.

How is the growth of the *hns* mutant? Is it impaired?

Growth of the *hns* mutant was slower but not significantly impaired. Growth defects in *hns* mutant backgrounds is often the case in *E. coli* however *C. rodentium* contains multiple orthologues of *hns* (we mutated the closest relative) which may account for the only mild effects on growth. We feel it unnecessary to include this as a figure in the text but have attached it here for your information.

The use of only Esp nomenclature can cause confusion. I figured EspI is NleA, adding this in parenthesis would be nice. Specially because NleA is regulated by Ler, and it would make it easier to correlate with literature that refers to this gene as *nleA*. People who are interested in NleA would miss this. I assume EspS has been described for the first time here, please clarify. This is actually a pretty cool point, because Citro causes hyperplasia and EHEC and EPEC don't. If the other two Esps also have an *nle* name please add it. These things help to the correlation with genomics papers and the Tobe PNAS 2006 *nle* paper.

We agree with the reviewer that nomenclature regarding effectors is difficult to clarify. Unfortunately, there is no generalized method of referring to all effectors and it is complicated by the fact that the repertoire of effectors differs between strains. To avoid arbitrary decisions, we have chosen to stick to the *C. rodentium* literature as much as possible. We have used the *C. rodentium* annotated genome to name effectors for the RNA-seq data. For the effectors discussed specifically, *EspS* has been discussed briefly previously (Petty *et al.*, 2010. J Bact 192(2) 525-38; Deng *et al.*, 2012. JBC 285(9) 6790-6800) and was referred to as *OspB* in the latter. We have chosen to use *EspS* so as to not create more confusion regarding the nomenclature. Only *EspI* has a more common alternative (*NleA*). As requested, we have included this in parentheses (referred to as *espI^{nleA}*) for any readers that are not familiar with *C. rodentium*.

Reviewer #2 (Remarks to the Author):

In the present study, the authors established a method to globally profile gene expression of the murine pathogen *Citrobacter rodentium* during infection of an in-vivo mouse model. Particularly, they isolate total RNA from the caecum and rectum of infected mice, i.e. two preferred *Citrobacter* niches

in vivo, at three distinct time points, enrich for bacterial mRNA, convert them into cDNA libraries, and sequence them. This allows the authors to follow the expression of *Citrobacter* virulence genes, such as locus of enterocyte effacement (LEE) and non-LEE (NLE) genes, over time. Deletion of one previously largely uncharacterized NLE effector (EspS), that was highly expressed at all time points and at both tissues, led to increased crypt lengths in the infected intestine as compared to wild-type-infected mice, although the molecular mechanism remains elusive. Besides, the authors identify genes involved in *Citrobacter* 1,2-propanediol (1,2-PD) metabolism to be strongly expressed during the peak of infection and uncover a link between 1,2-PD and LEE expression. Interestingly, not 1,2-PD itself, but its conversion into propionate and sensing of the latter was responsible for LEE induction, and deletion of the pdu operon, encoding genes required for 1,2-PD to propionate conversion, facilitates *Citrobacter* clearance by the infected host.

Major comments:

Reviewing this manuscript, I was specifically asked to evaluate the RNA-seq-related parts of the study. Sampling, RNA isolation, bacterial mRNA enrichment (MicrobEnrich, MicroExpress), cDNA library generation and the sequencing protocol are state of the art. However, I was surprised by the tiny fraction of reads that mapped to *Citrobacter* mRNAs (typically <0.1% of the total reads in the in-vivo samples). As the samples were sequenced to ~35 Mio total reads per library, the fraction of “useful” reads (mapping to *Citrobacter* ORFs) is in the range of 1,000-80,000 reads. This is extremely little! How do the authors explain that? Was the enrichment of bacterial mRNAs not efficient? What are the remaining 99.9% of the reads? Do they map against the murine genome, against *Citrobacter* sequences (but to multiple loci, and thus got discarded as the protocol selects for uniquely mapped reads), or against the genome of other (non-*Citrobacter*) bacterial species (members of the mouse microbiota)? In general, it is highly recommended that the authors include a separate supplementary table (or expand dataset S1) to include seq. depth and mapping stats for all samples (input reads, quality-filtered reads, mappable reads, uniquely mapped reads, etc. for all libraries).

While we agree with the reviewer that the percentage of reads mapping uniquely to *C. rodentium* obtained from complex samples as described here is small we don't find it all that surprising. According similar studies using *in vivo* RNA-seq approaches, ranges of mapped reads to non-ribosomal regions of reference genomes include ~0.05-2.5% (Szafranska *et al.*, 2014. nMbio 5 (6) e01775-14), ~0.0017-0.0022 (Avican *et al.*, 2015. PLOS Path 11(1): e1004600) and ~0.5% (Nuss *et al.*, 2017. PNAS 114 (5) E791-E800), which is not so different to the ~0.004-0.23% obtained in our study. In our experience, enrichment of mRNA is an inefficient process even when using pure culture samples. Given that the samples we used were from biopsies, we were not able to control the number of cells returned per sample and thus could not determine the most appropriate concentration of total RNA to run through the enrichment procedure in advance, which is a limiting step in the efficiency of these protocols. Furthermore, performing multiple enrichments would have resulted in loss of total yield at each clean-up stage, a risk we could not take due to limited samples, complicating matters even more. However, despite these limitations and the depth of sequencing that we could obtain from the platform we had available, we felt as if we had enough replicates and timepoints sampled to obtain useable and robust data. We were also very pleased with the correlation between replicate infections giving us confidence in the data. As requested, we have clarified this point by removing figures summarising % mapping to ORFs of *C. rodentium* and instead expanded Dataset 1 to contain more detailed information of the % reads that relate to *C. rodentium*, the host, unmapped etc.

Differential expression analysis is performed using EdgeR, which is state of the art (although I was surprised by the number of significantly differentially expressed genes called by EdgeR (around 100 per comparison), given the low sequence coverage of the in-vivo samples). In general, however, the (differential) expression data is presented in a rather confusing fashion: It is unclear to this reviewer why sometimes RPKMO-values are plotted and sometimes EdgeR-calculated fold-changes, and I fear that it will be even more confusing for readers less familiar with RNA-seq. Given that the authors anyways filter out lowly expressed/unexpressed genes and EdgeR considers abundance of a transcript for differential expression calling, why don't the authors just entirely rely on the EdgeR fold-changes and plot these throughout the manuscript?

We agree with the reviewer on this point. We calculated RPKMO values to generate heatmaps and visualise expression patterns of genes that are perhaps not significantly differentially expressed (as determined by EdgeR fold changes) – for instance to look at LEE expression dynamics. However, this may not be clear and can confuse the question being asked. We have therefore discarded these

values and generated new figures where appropriate replacing RPKMO expression values for EdgeR derived fold changes in all cases. We agree that this is much more straight forward and clear in the eyes of the reader.

The figures are dominated by heatmaps for visualizing expression data, which is not ideal. Since the authors have time-course data, I'd strongly recommend they select for one in-vitro reference control, and convert their heatmaps into line plots (reference → day 3 → day 7 → day 10; each one diagram for the respective tissue). Also, plotting the expression kinetics of all genes in light grey and highlighting the ones referred to in the main text in color on top, would allow the reader to put expression patterns of e.g. LEE and NLE genes, or the pdu operon, in context with the global expression changes during the time-course of infection.

In agreement with the above comment we have remade figures throughout the manuscript to use only EdgeR derived values for clarity. We feel the heatmaps of LEE and NLE expression dynamics are of high value to researchers who work in this area but not to non-experts from a wider audience. We have therefore taken the reviewers advice and converted these to heatmaps generated from EdgeR data and moved them to Supplementary data. In replace of them, we have generated simpler line graphs to highlight LEE and NLE expression dynamics over the time course of infection. We have not plotted all genes in light grey however given the lower resolution nature of the data and the fact that genes may not be detected at all timepoints. We agree that this offers a much clearer way of presenting the main findings, while experts can still visualise the full heatmaps in the supplementary data.

Minor comments:

Wording: RNA-seq measures the steady-state levels of cellular RNAs, i.e. the sum of transcription AND decay. In contrast, the authors speak about "transcriptional landscape", "transcriptional activity" and similar throughout their manuscript (examples include, but are not restricted to, lanes 139, 415, 473, 483, 855, 870, 878, 881). This should be strictly avoided and replaced by terms such as "transcript levels", "expression" (or similar).

This has been corrected throughout the manuscript.

What makes it difficult to follow the flow of the story is that the main text sometimes appears to refer to the wrong figures/panels. Examples are lane 194 (Fig. S2B must be Fig. S3A), lane 212 (Fig. S3C must be Fig. 3C), lanes 865 + 867 (panels E and F must be panels C and D), lane 950 (6E is probably rather be 6C).

We apologise for any confusion caused to the reviewers. The figures have been thoroughly amended and unnecessary panels removed, which we feel simplifies the paper and makes it more digestible. We hope that the corrected text and figures are now satisfactory.

There should be more information in the main text of the manuscript describing the sampling of the RNA, given that it represents the major dataset of the study. E.g. it is not mentioned in the main text how many replicates were taken and analyzed.

We have added a shortened description of the method to the main text, highlighting both the method of tissue isolation and number of replicates used.

Lane 143 and following: How much % of these were of Citrobacter origin (what read number refers to 100% in Suppl. Fig. S2B)? Only mentioning that each library was sequenced to 26-45 Mio reads (lane 143) and that ~25% of the Citrobacter reads were uniquely mapped to ORFs (lane 146) is highly misleading as it implies a high Citrobacter sequence coverage, which is not the case as mentioned above.

We agree that this is misleading and have removed both the statements and the figures for clarity on the issue. As above, Dataset 1 has been amended to simplify the presentation of the data.

Lane 160: It's true that *regA* mRNA was detected, but it was not induced relative to the in-vitro samples (unlike *kfcC*). What's known about RegA-mediated activation? E.g. is RegA activated at the protein level, or by phosphorylation? Please give more information here.

RegA is constitutively expressed and the protein is functionally responsive to bicarbonate, binding this and activating transcription of downstream genes in response to this signal. For this reason, we would not expect *regA* levels to be dramatically affected. We have clarified this point in the text and added the reference Tan *et al.*, 2011. *J Bact* 193(7): 1777–1782.

Lane 183: I am not sure if the abbreviation "NLE" is ever explained in the text.

Thank you for pointing this out. We have added the description where NLE first appears in the text.

Lane 213: This sentence suggests that there is qRT-PCR data for *escL* and *escV* in Fig. S3D, which I couldn't find (neither in Fig. S3D nor elsewhere).

The qPCR data for *LEE1* and *LEE3* did not examine these genes. We have clarified this in the text.

Figure 2A occupies a lot of space but contains rather limited information. A simple line plot (or heatmap) showing expression of these genes would do. The read coverage plot might be moved to the supplementary material instead.

We agree with this however we would like to show a smaller version of the graph to illustrate the global pattern of gene expression *in vivo*. In order to make this possible, we have dramatically condensed Figures 1 to 3, removing unnecessary panels as requested by the reviewer. This has created a much more concise presentation of the data. We have combined the read coverage plot at peak infection with the *LEE* expression dynamics data to create a new figure 2 that focuses on virulence factors as described in the text. We feel this is much more informative and space efficient.

Figure 2B: This table can be moved to the supplementary material, as the numbers of DEGs may be inferred from panels C + D.

This has been removed.

Figure 2C: Please check numbers again. E.g. caecum, day 3: 32 DEGs in panel B, 33 in panel C, 32 in panel D.

We have corrected the numbers throughout the manuscript.

Figure 3: It is often unclear (and for the reader very confusing!) why sometimes RPKMO-values and sometimes fold-changes in gene expression are plotted. I'd strongly recommend to stick to one way of visualizing gene expression (preferentially, chose one of the 2 in-vitro conditions as a reference and display the EdgeR-calculated gene expression changes throughout the study).

Yes, we agree with this and have made dramatic changes to accommodate this point, as detailed above in the major concerns section.

Figure 3A: Are the data in the heatmap representative for 3 replicates, or does it display the mean expression from all 3 biol. replicates. This is an important difference! While it was fine to show a representative image of the read coverage in Fig. 2A, due to space limitations), there is no reason for

why not to plot the mean from all 3 replicates in this heatmap (rather than 1 representative replicate). Please clarify/change accordingly (same goes for Fig. 4A).

Yes, all RNA-seq data presentation in figures (except for the coverage graphs) is representative of the mean of all replicates and statistically robust data. We have clarified this in all relevant figure legends to avoid any confusion.

Figure 3B: What does "undetected" in the x-axis label mean? Doesn't the materials & methods section say that these genes were filtered out?! In general, this panel is little informative and may be moved to the supplementary material.

This was meant to describe ORFs that perhaps fell below cutoff for read mapping due to sequencing resolution rather than merely downregulation. However, we agree that this is confusing and have removed this figure from the manuscript.

Figure 3C: The legend says that LEE operons "that were differentially expressed" are shown. Differentially expressed compared to what? The legend further says that the genes *escL*, *escD* and *escV* are representative examples. Again: representative for what? Please, be more specific. Similar examples include the legend to Fig. 4B and Fig. 5B.

All *in vivo* RNA-seq data was compared to growth in DMEM but for clarity we have made this more apparent in all appropriate figure legends. The figure panels 4B and 5B mentioned above have subsequently been removed as they plotted RPKMO values, which are no longer a part of the text.

Figure 3D: In the opinion of this reviewer, this is not worth a dedicated model figure (or at least, it should be moved to the supplementary material).

This has been removed entirely.

Figure 4: For NLE genes for which the regulator was not identified, wouldn't the global, temporal expression data be predestined to perform a correlation analysis (of the respective effector with known members of global regulons) to predict those regulators?

This is a nice suggestion from the reviewer and could indeed be investigated in the future but we feel it is not required to bolster the hypothesis presented in this work and may just be adding data that is not strictly required.

Fig. 5B: This doesn't look like high fold-changes for many genes (close to 0, according to the color-code), even though according to the description the top-induced genes were plotted. Please clarify (and better explain in the legend or modify figure accordingly; same goes for Supplementary Figure S5B). In general (instead of panels B +C) it would be more intuitive to show 2 line plot diagrams (1 for caecum, 1 for rectum), with expression changes of all genes over the time-course as grey lines and highlight the *pdu* genes in colors. This way, the reader can much better put the expression of the respective genes into context).

The reason for the color coding not representing the impact of the fold changes was due to expression fold changes of *kfcC* being so much higher than other genes, even on the log₂ scale. We have therefore removed these heatmaps (as requested above) and replaced the figure with a much simpler bar chart highlighted the top fold changes, including members of the *pdu* operon.

Figure 5E: Please explain the abbreviations (e.g. "PD", "BMC").

This has been corrected.

Figure 6E: The WT strain should be included as a reference.

This has been included.

Figure 6F: It's stated nowhere how many replicates were analyzed.

It was stated in the methods section but has now also been mentioned in the main text and the figure legend.

Supplementary Figure S3A: Why not show most-INDUCED genes compared to in-vitro (rather than most ABUNDANT)?

This data has been removed from the Supplementary material as the information can be obtained from Datasets 1 and 2.

Supplementary Figure S5A is little informative. Is it really needed?

We added this figure to the supplementary for completeness. Rather than remove it, we have labelled it more informatively.

Reviewer #3 (Remarks to the Author):

In this work, the authors investigated host-derived signals that are used as cues by *Citrobacter rodentium* to help regulate bacterial gene expression required for adaptation to the host environment. They found that the metabolism of 1,2-PD derived from the microbiota was involved in a signaling/sensing system that coincided with the expression of the T3SS and associated effector proteins. The authors speculate that the metabolism of microbiota-derived 1,2-PD signals directly to *Citrobacter* in order to increase fitness in the host environment through the effects it has on bacterial T3SS gene expression.

Below are some areas that the authors may wish to consider in order to strengthen the findings and conclusions.

For the in vivo transcriptional profiling of CR, the method hinges on the ability to detect transcripts derived specifically from CR. However, given the high degree of orthologous genes between CR and other Enterobacteriaceae, how can the authors be sure that these RNA signals are derived from CR and not other members of the microbiota? While I understand that LEE genes would not be present in the normal microbiota, what does "high stringency" read mapping actually mean and how does it exclude non-CR RNAs from orthologous genes that are not unique to CR? This is particularly important in the later part of the manuscript when highly common genes involved in metabolic flux are introduced.

This is an important point raised and one that is always going to be a challenge with this type of approach. We will try to clarify these concerns here. The first step taken in the experiment was to gently clear the luminal contents of the tissue without causing disruption to the epithelial surface (where *C. rodentium* would be attached and localized in relatively high numbers). While this cannot guarantee removal of commensal bacterial RNA it acts as an enrichment-like step. Secondly, the mapping was performed using more strict mapping criteria than would normally be used by default parameters (this was why we described it as high stringency) to try and add another level of specificity to the procedure. Thirdly, we used uninfected mice processed in the same manner as the infected tissue as a control. We reasoned that this would give us a good idea of how reads derived from the residual microbiota may map to the *C. rodentium* genome sequence. Nearly all of the cross mapping from uninfected samples related to ribosomal/transfer RNA reads with very few reads mapping to

ORFs to be considered as *C. rodentium*, as described in the supplementary data and methods sections. For this reason, we performed the analysis based entirely on reads that mapped uniquely to *C. rodentium* ORFs. As the reviewer pointed out, key virulence genes of interest would only be found in *C. rodentium*, as are the *pdu* genes in this case. We were confident that other metabolic genes of interest were of *C. rodentium* origin given that uninfected samples did not cross map significantly and that the % of commensal strain abundance would be reduced during infection, further enriching for *C. rodentium*.

Figure 1 – the colonization dynamics of CR in resistant mice is well known, as is the location of colonization over time. This figure does not add anything to the literature that is not already known. I would be more interested (or concerned) with whether the constitutive expression of luciferase changes the tissue specificity, bacterial density, inflammation, or clearance rates of the CR compared to the uncontrived wild-type bacteria, as it is known that such reporters can alter fitness etc *in vivo*.

The use of luminescent *C. rodentium* for murine infection studies is well established and has been used by us and many other groups extensively. We have never encountered any aberrant effects of LUX marked bacteria in comparison to infection with non-modified WT bacteria. A recent study has confirmed the use of this model by phenotypically comparing LUX marked bacteria to WT cells, with no-metabolic disadvantage of the former (Read *et al.*, 2016. PeerJ 4: e2130). It was also concluded that in single infections, like we have performed here, marked *C. rodentium* infection is virtually indistinguishable from that of WT. We would like to maintain this figure as it shows the reader (and in particular non-experts) the infectious state of each particular mouse (and associated tissue) before transcriptome analysis, highlighting the workflow of the experiment. However, in attempt to be more economical with space (as discussed above for reviewer 2) we have merged Figure 1 with Figure 2, eliminating unnecessary panels from the figures.

The idea that the time when bacteria are sampled *in vivo* would alter their transcriptional profile is not all that surprising. Would it not make sense to try and correlate these different gene expression signatures with the host response and/or bacterial density, which are both changing dramatically over 3-10 days post infection?

While it is true that one would expect infection-specific gene expression changes, this has not been described for *C. rodentium* and only on a few recent occasions for other organisms (see examples as cited above in response to reviewer 2). Focusing on the bacterial side of the study therefore provides valuable information to the field. While we agree it would be interesting to perform a dual-RNA-seq approach in this context, we feel this would convolute the study too much and would be more suited to investigation independently. As discussed in our manuscript, we were most interested in learning of pathogen virulence-metabolic interplay during infection, which we feel like we achieved successfully.

The narrative around the dynamics of LEE gene expression *in vivo* (Figure 3) – in part C, without error bars or a statistical analysis I'm not sure I would agree that *escL* and *escV* "decrease" over time; nor is the biological significance of this understood, as the bacteria are presumably replicating and being cleared over this time period in a highly dynamic fashion.

On reflection we agree with the reviewer on this point and would like to refer them to similar points raised by reviewer 2. The data were indeed generated from triplicate data, with significant changes in expression discussed however we have removed the RPKMO generated data (which was used to assess LEE dynamics) and instead focused on the EdgeR derived data, and statistically significant fold changes in expression.

Without being able to complement your *espS* mutant regarding crypt length phenotypes, I would be cautious in saying that *EspS* represses hyperplasia *in vivo*. There could be numerous other explanations for this phenotype. It is not clear even what this line of investigation has to do with the question being addressed in the manuscript.

This line of investigation was a sidestep from the gene expression work but we strongly feel it complements the discovery appropriately by offering physiological insight into the discovery of *EspS* expression *in vivo*. Furthermore, as highlighted kindly by reviewer 1, the fact that *EspS* is specific to *C. rodentium* and appears to mediate a *C. rodentium*-specific phenotypic hallmark of infection is potentially very informative. While we agree complementation would have been ideal, we were

satisfied with the phenotype given the fact that the experiment was performed on two entirely independent occasions with statistically robust data generated each time. However, to avoid any unwarranted claims we have clarified in the text that we only hypothesize EspS may play a role in repression of hyperplasia and that further studies would be required to elucidate a precise mechanism.

What is the 1,2-PD concentration *in vivo* in your experimental conditions? With 10 mM leading to a ~2-fold increase in gene expression *in vitro*, I'm concerned about the biological significance of this potential signaling pathway. Have the authors considered previously published work showing that propionate inhibits T3SS gene expression in *Salmonella* (Lawhon, 2002, *Mol Micro*), which is an opposite effect to that claimed here for CR?

We refer the reviewer to discussion above on 1,2-PD concentrations *in vivo*, as this point was also raised by reviewer 1. The concentrations of 1 mM leading to expression changes of 2-fold is not all that surprising. The LEE is a hub of complex regulation involving numerous signals that converge on one element, many of which likely have a degree of redundancy and merely contribute to the infection process (Connolly *et al.*, 2015. *Front Microbiol* 6:568). We would therefore not expect all possible inputs to result in large fold changes in gene expression, particularly when the system is already active in a synthetic setting *in vitro*. However, we feel strongly that the combination of our *in vitro* mechanistic studies with the *in vivo* relevance of the discovery (in addition to our extended data on inducing concentrations of 1,2-PD) support the claims that 1,2-PD at least contributes to niche specificity of this organism. In regard to the comment on repression T3SS expression by propionate in *Salmonella*, we do not dispute this but would like to reiterate the fact that it is known SCFAs (including propionate) have the opposite effect on LEE expression in EHEC (Nakanishi *et al.*, 2009. *Microbiology* 155(2):521-30) and as such *C. rodentium* as we have demonstrated here.

The fact that a CR mutant unable to metabolize 1,2-PD had no colonization defect at the peak of infection (when T3SS would be most needed and most active), seems to contradict your conclusion that this secondary metabolite is needed to activate T3SS. What happens to T3SS gene expression *in vivo* when you use this pduA-J mutant? According to your hypothesis, T3SS gene expression should be reduced, as would colonization.

To clarify, we are not suggesting that this signal is absolutely required to activate T3SS but rather it plays an important role in modulation of the system and contributes to the overall dynamic of infection. As we mention in the text, it is possible that the localized metabolic makeup of the gut may change over time during infection and thus results in a strong phenotype towards the latter stages of infection. However, this would be extremely difficult to investigate and extends beyond the scope of the current study. Alternatively, we have attempted to specifically address the question of T3SS regulation by 1,2-PD *in vivo*. As described for reviewer 1, we successfully reversed the colonization defect of the pdu mutant by constitutively expressing Ler from its native chromosomal location. This result demonstrates that we can overcome the colonization defect by constitutively expressing the T3SS, proving that it is the effect of 1,2-PD metabolism on regulation and not a fitness defect that accounts for the phenotype.

Reviewers' comments:

Reviewer #1 (Remarks to the Author):

I really liked this manuscript the first time around, and think this is very novel and an important contribution to the field. I asked for some extra experiments and controls to tie some loose ends, and they were all performed properly. All of my points were addressed, and I think this is an important study with broad implications.

Reviewer #2 (Remarks to the Author):

The authors satisfactorily addressed my comments. Well done!

Reviewer #3 (Remarks to the Author):

The authors have addressed the comments of the reviewers only partially. I was surprised that the authors were dismissive of the request to complement their mutants, which is standard microbiology practice. Citing another paper where similar mutants were complemented seems to miss the point entirely of why one would complement a mutant (to demonstrate the absence of unlinked mutations in their particular strains).

The important question of biological significance regarding 1,2 PD concentration in mice with a complex microbiota was not addressed. The authors chose to cite a relatively new paper where another group measured 1,2 PD in mice that were previously germ-free and mono-associated with *B. theta* (where [1,2 PD] reached ~ 40 µM), which they say would be similar or higher in SPF mice. Given that the conclusions of the paper rest on this information, I think this is much too speculative. In mono-associated mice, *B. theta* reaches stable densities of 10¹⁰ cfu/g feces, whereas *Bacteriodes* species do not colonize SPF mice even after inoculation at very high doses (Lee SM, 2013 Nature). Therefore, one can't simply extrapolate mono-associated mice to SPF mice. The fact that, in vitro, a concentration range of 1,2 PD of 0.01-0.1 mM produces essentially no dose response on LEE gene expression (Figure S6b) makes it hard to reconcile their claim that 1,2 PD is a chemical signal that 'modulates' LEE gene expression and type III-dependent colonization in the gut.

Finally, an explanation for why a *Citrobacter* mutant that can't use 1,2 PD has no colonization defect compared to wild type between the time of infection and the peak bacterial load (~day 10) has not been addressed. Between day 1-10 is when type 3-dependent colonization is most important. After day 10, host clearance processes begin and so one should be careful about interpreting differences here as strictly colonization differences. The explanation that 1,2 PD is involved in "modulation" of the (type III) system in vivo has not been formally demonstrated for a time point after peak colonization, nor have the authors defined what 'modulation' actually means. This is important because their RNA seq analysis from infected animals was done at time points when the pdu mutant has no colonization defect.

Reviewer #1

I really liked this manuscript the first time around, and think this is very novel and an important contribution to the field. I asked for some extra experiments and controls to tie some loose ends, and they were all performed properly. All of my points were addressed, and I think this is an important study with broad implications.

Reviewer #2

The authors satisfactorily addressed my comments. Well done!

It was satisfying to read that reviewers 1 and 2 felt the additional experiments and controls helped address all their concerns. We would like to thank them for their constructive criticism that has helped to improve our manuscript. The third reviewer felt there were some areas of the paper that needed strengthening. We have modified the paper where appropriate as outlined below.

Reviewer #3

The authors have addressed the comments of the reviewers only partially. I was surprised that the authors were dismissive of the request to complement their mutants, which is standard microbiology practice. Citing another paper where similar mutants were complemented seems to miss the point entirely of why one would complement a mutant (to demonstrate the absence of unlinked mutations in their particular strains).

We fully understand the need to complement mutants. Most of our paper is focused on the *pdu* operon where we performed the expected complementation experiments. The reviewer is referring to the data presented in one panel of a supplementary figure (Fig S4e) where we use four regulatory deletion strains to show that each of the effectors we studied have distinct patterns of control, which was successfully demonstrated. We have now repeated these experiments to include complementation. Accordingly, we have added this data to Figure S4e and the appropriate text in the manuscript to reflect this successful complementation.

The important question of biological significance regarding 1,2 PD concentration in mice with a complex microbiota was not addressed. The authors chose to cite a relatively new paper where another group measured 1,2 PD in mice that were previously germ-free and mono-associated with *B. theta* (where [1,2 PD] reached ~ 40 microM), which they say would be similar or higher in SPF mice. Given that the conclusions of the paper rest on this information, I think this is much too speculative. In mono-associated mice, *B. theta* reaches stable densities of 10e10 cfu/g feces, whereas *Bacteriodes* species do not colonize SPF mice even after inoculation at very high doses (Lee SM, 2013 Nature). Therefore, one can't simply extrapolate mono-associated mice to SPF mice. The fact that, in vitro, a concentration range of 1,2 PD of 0.01-0.1 mM produces essentially no dose response on LEE gene expression (Figure S6b) makes it hard to reconcile their claim that 1,2 PD is a chemical signal that 'modulates' LEE gene expression and type III-dependent colonization in the gut.

We agree it would be interesting to determine the precise concentration of 1,2PD in mice with a complete microbiota but we have been unable to perform this experiment. There are two reasons. Firstly, we do not have access to the necessary equipment, namely, a highly sensitive GCMS coupled with precise labeled standards as internal controls. Secondly, we think it would be extremely difficult to accurately measure mM concentrations of 1,2PD in complex samples obtained from the gut lumen. Given this limitation, we cited the one study where this metabolite has been measured, which was in mice mono-associated with *B. theta*. In that study, they measured a concentration of around 40 mM, which is interesting because we see that around 50 mM causes an increase in expression of the T3SS *in vitro*. It therefore seems appropriate to cite and explain their data, despite the limitations. Finally, our new data in Figure 6b confirms the role of type 3 mediated virulence regulation in relation to 1,2PD metabolism *in vivo*. Hence, it is reasonable to assume that mM concentrations of 1,2PD are likely present in the GI tract of mice with a complex microbiota.

Finally, an explanation for why a *Citrobacter* mutant that can't use 1,2 PD has no colonization defect compared to wild type between the time of infection and the peak bacterial load (~day 10) has not been addressed. Between day 1-10 is when type 3-dependent colonization is most important. After day 10, host clearance processes begin and so one should be careful about interpreting differences here as strictly colonization differences. The explanation that 1,2 PD is involved in "modulation" of the (type III) system *in vivo* has not been formally demonstrated for a time point after peak colonization, nor have the authors defined what 'modulation' actually means. This is important because their RNA seq analysis from infected animals was done at time points when the pdu mutant has no colonization defect.

The main focus of this work was to identify *in vivo* expressed genes and show their relevance during an infection. Using our model system we demonstrate that the *pdu* operon is important for the persistence of *Citrobacter* and that the mechanism lies through changes in LEE expression. Clearly our data raises further questions, such as why differences are seen after the peak of the infection has been reached. Whilst it is exciting our data raises new questions, addressing them would warrant separate studies that are beyond the scope of the current work.

We can of course speculate as to why there is no colonization defect at early time points. It is known that expression of the T3SS is subject to many different regulatory inputs and affected by numerous environmental signals. One possibility is that, early in the infection process, the LEE is strongly regulated by factors such as pH (Puhar et al., 2014), changes in oxygen tension (Lopez et al., 2016) or short chain fatty acids (Nakanishi et al., 2009) that act as key signals to activate the production of the T3SS. However, as the *Citrobacter* infection becomes established and the population expands then other environmental factors including 1,2 PD help fine tune the control of virulence and aid in persistence. This would be a logical evolutionary adaptation to ensure better colonization of the host. We have added some lines in the discussion to address this point:

"Furthermore, this mechanism appeared to be crucial at the latter stages of the host colonization suggesting a dynamic response to the nutritional status of the niche at all stages of the infection. This demonstrates an intuitive mechanism whereby the pathogen generates its own positive cues for virulence depending on the metabolic status of the host."

We very much hope the above changes are satisfactory and the manuscript is now suitable for publication.